# Unsupervised Representation Learning of Brain Activity via Bridging Voxel Activity and Functional Connectivity

**Ali Behrouz**[*]
Cornell University
ab2947@cornell.edu

**Parsa Delavari**[*]
University of British Columbia
parsadlr@student.ubc.ca

**Farnoosh Hashemi**[*]
Cornell University
sh2574@cornell.edu

## Abstract

Effective brain representation learning is a key step toward the understanding of cognitive processes and unlocking detecting and potential therapeutic interventions for neurological diseases/disorders. Existing studies have focused on either (1) voxel-level activity, where only a single weight relating the voxel activity to the task (i.e., aggregation of voxel activity over a time window) is considered, missing their temporal dynamics, or (2) functional connectivity of the brain in the level of region of interests, missing voxel-level activities. In this paper, we bridge this gap and design BRAINMIXER, an unsupervised learning framework that effectively utilizes both functional connectivity and associated time series of voxels to learn voxel-level representation in an unsupervised manner. BRAINMIXER employs two simple yet effective MLP-based encoders to simultaneously learn the dynamics of voxel-level signals and their functional correlations. To encode voxel activity, BRAINMIXERfuses information across both time and voxel dimensions via a dynamic self-attention mechanism. To learn the structure of the functional connectivity graph, BRAINMIXER presents a temporal graph patching and encodes each patch by combining its nodes' features via a new adaptive temporal pooling. Our experiments show that BRAINMIXER attains outstanding performance and outperforms 14 baselines in different downstream tasks and experimental setups.

## 1 Introduction

The recent advancement of neuroimaging provide rich information to analyze the human brain. The provided data, however, is high-dimensional and complex [68], which makes it hard to take advantage of powerful machine learning models in analyzing them. To overcome this challenge, representation learning serves as the backbone of machine learning methods on neuroimage data and provides a low-dimensional representation of brain components at different levels of granularity, enabling the understanding of behaviors [77], brain functions [97] and/or detecting neurological diseases [84].

In the brain imaging literature, studies have mainly focused on two spatial scales—voxel-level and network-level—as well as two analysis approaches—multivariate pattern analysis (MVPA) and functional connectivity [60, 85]. The MVPA approach is often employed at the voxel-level scale and in task-based studies to associate neural activities at a very fine-grained and local level with particular cognitive functions, behaviors, or stimuli. This method has found applications in various areas, including the detection of neurological conditions [81, 11], neurofeedback interventions [22], decoding neural responses to visual stimuli [41], deciphering memory contents [53, 15], and classifying cognitive states [64]. The functional connectivity analysis, on the other hand, focuses on the temporal correlations or statistical dependencies between the activity of different brain regions at

---

[*]Equal Contribution.

NeurIPS 2023 AI for Science Workshop.

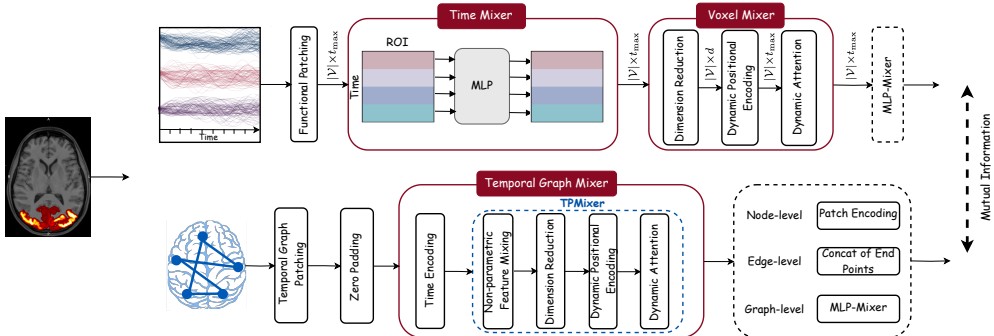

Figure 1: **Schematic of the BRAINMIXER**. BRAINMIXER consists of two main modules: (1) Voxel Activity Encoder (top), and (2) Functional Connectivity Encoder (bottom).

larger scales to assess how these areas communicate and collaborate. This method has been utilized to study various topics such as task-related network dynamics [32, 43] and the effects of neurological disorders on brain connectivity [33, 29].

**Limitation of Previous Methods.** Despite the advances in the representation learning of brain signals, existing studies suffer from a subset of five limitations: ① Study the human brain at a single scale: Most existing studies study the brain at either voxel-level or functional connectivity, while these two scales can provide complementary information to each other; e.g., although voxel-level activity provides detailed and more accurate information about brain activity, it misses the information about how different areas communicate with each other at a high level. Recently, this limitation has motivated researchers to search for new methods of integrating these two levels of analyses [66, 62]. ② Supervised setting: Learning brain activity in a supervised setting relies on a large number of clinical labels while obtaining accurate and reliable clinical labels is challenging due to its high cost [3].③ Missing information by averaging: Most existing studies on voxel activities aggregate measured voxel activity (e.g., its blood-oxygen level dependence) over each time window to obtain a single beta weight [73, 88, 72]. However, this approach misses the voxel activity dynamic over each task. Moreover, most studies on brain functional connectivity also aggregate closed voxels to obtain brain activity in the Region of Interest (ROI) level, missing individual voxel activities. ④ Missing the dynamics of the interactions: Some existing studies neglect the fact that the functional connectivity of the human brain dynamically changes over time, even in resting-state neuroimaging data [14]. In task-dependent neuroimage data, subjects are asked to perform different tasks in different time windows, and the dynamics of the brain activity play an important role in understanding neurological disease/disorder [39]. ⑤ Designed for a particular task or neuroimaging modality: Due to the different and complex clinical patterns of brain signals [27], some existing methods are designed for a particular type of brain signal data [52, 13], and there is a lack of a unified framework.

**Application to Understanding Object Representation in the Brain.** Understanding object representation in the brain is a key step toward revealing the basic building blocks of human visual processing [37]. Due to the hierarchical nature of human visual processing, it requires analyzing brain activity at different scales, i.e., both functional connectivity and voxel activity. However, there is a small number of studies in this area, possibly due to the lack of proper large-scale datasets. Recently, Hebart et al. [37] provided a large-scale fMRI and MEG datasets, THINGS, to fill this gap. However, the preprocessed data by Hebart et al. [37] not only does not provide functional connectivity, but it also has aggregated voxel activity over each time window, and provides a single beta weight for each voxel, missing dynamics of voxel activity. To address this limitation, we present two newly preprocessed versions of this dataset that provide both functional connectivity and voxel activity timeseries of fMRI and MEG modalities. See Appendix B for more details.

**Contributions.** To overcome the above limitations, we leverage both voxel-level activity and functional connectivity of the brain. We present BRAINMIXER, an unsupervised MLP-based brain representation learning approach that jointly learns representations of the voxel activity and functional connectivity. BRAINMIXER uses a novel multivariate timeseries encoder that binds information across both time and voxel dimensions. It uses a simple MLP with functional patching to fuse information across different timestamps and learns dynamic self-attention weights to fuse information across voxels based on their functionality. On the other hand, BRAINMIXER uses a novel temporal graph

learning method to encode the brain functional connectivity. The graph encoder first extracts temporal patches using temporal random walks and then fuses information within each patch using the designed dynamic self-attention mechanism. We further propose an adaptive permutation invariant pooling to obtain patch encodings. Since voxel activity and functional connectivity encodings are different views of the same context, we propose an unsupervised pre-training approach to jointly learn voxel activity and functional connectivity by maximizing their mutual information. In the experimental evaluations, we provide two new large-scale graph and timeseries datasets based on THINGS [37]. Extensive experiments on six datasets show the superior performance of BRAINMIXER and the significance of each of its components in a variety of downstream tasks.

For the sake of consistency, we explain BRAINMIXER for fMRI modality; however, as it is shown in §4, it can simply be used for any other neuroimaging modalities that provide a timeseries for each part of the brain (e.g., MEG and EEG). When dealing with MEG or EEG, we can replace the term "voxel" with "channel". Supplementary materials can be found in this link.

## 2   Related Work

**Timeseries Learning.** Attention mechanisms are powerful models to capture long-range dependencies and so recently, Transformer-based models have attracted much attention in time series forecasting [103, 55]. Due to the quadratic time complexity of attention mechanisms, several studies aim to reduce the time and memory usage of these methods [20]. Another type of work uses (hyper)graph learning frameworks to learn (higher-order) patterns in timeseries [67, 75]. Inspired by the recent success of MLP-MIXER [82], Li et al. [57] and Chen et al. [19] presented two variants of MLP-MIXER for timeseries forecasting. All these methods are different from BRAINMIXER, as ① they use static attention mechanisms, ② do not take advantage of the functionality of voxels in patching, and ③ are designed for timeseries forecasting and cannot simply be extended to various downstream tasks on the brain.

**MLP-based Graphs Learning.** Learning on graphs has been an active research area in recent years [44, 90, 16]. While most studies use message-passing frameworks to learn the local and global structure of the graph, recently, due to the success of MLP-based methods [82], MLP-based graph learning methods have attracted much attention [42, 10]. For example, Cong et al. [21] and He et al. [35] presented two extensions of MLP-MIXER to graph-structured data. However, all these methods are different from BRAINMIXER and specifically FC Encoder, as either ① use time-consuming graph clustering algorithms for patching, ② are static methods and cannot capture temporal properties, or ③ are attention-free and cannot capture the importance of nodes.

**Graph Learning and Timeseries for Neuroscience.** In recent years, several studies have analyzed functional connectivity to differentiate human brains with a neurological disease/disorder [45, 18, 93]. With the success of graph neural networks in graph data analysis, deep learning models have been developed to predict brain diseases by studying brain network structures [7, 108, 26]. Moreover, several studies focus on brain signals [24, 79] to detect neurological diseases. For example, Cai et al. [13] designed a self-supervised learning framework to detect seizures from EEG and SEEG data. However, these methods are different from BRAINMIXER as they are designed for a particular task (e.g., classification), a particular neuroimaging modality (e.g., fMRI or EEG), and/or supervised settings.

## 3   Method: BRAINMIXER

Detailed discussion about background concepts can be found in Appendix A.

**Notation.** We represent the neuroimaging of a human brain as $\mathcal{B} = \{\mathcal{B}^{(t)}\}_{t=1}^{T}$ where $\mathcal{B}^{(t)} = (\mathcal{V}, \mathcal{G}_F^{(t)}, \mathcal{X}^{(t)}, \mathbb{F})$ represents the neural data in time window $1 \leq t \leq T$. Here, $\mathcal{V}$ is the set of voxels, $\mathcal{G}_F^{(t)} = (\mathcal{V}, \mathcal{E}^{(t)}, \mathcal{A}^{(t)})$ is the functional connectivity graph, $\mathcal{E}^{(t)} \subseteq \mathcal{V} \times \mathcal{V}$ is the set connections between voxels, $\mathcal{A}^{(t)}$ is the correlation matrix (weighted adjacency matrix of $\mathcal{G}_F^{(t)}$), $\mathcal{X}^{(t)} \in \mathbb{R}^{|\mathcal{V}| \times \tilde{T}(t)}$ is a multivariate timeseries of voxels activities, $\tilde{T}(t)$ is the length of the timeseries, and $\mathbb{F}$ is the set of functional systems in the brain [76] in time window $t$. In task-dependent data, each time window $t$ corresponds to a task, and in resting state data, we have $T = 1$. We let $t_{\max} = \max_{t=1,\dots,T} \tilde{T}(t)$,

representing the maximum length of timeseries. BRAINMIXER consists of two main modules ①
*Voxel Activity* (VA) Encoder and ② Functional Connectivity (FC) Encoder:

## 3.1 Voxel Activity Encoder

The main goal of this module is to learn the time series of the voxel-level activity. However, the
activities of voxels are not disjoint; for example, an increase in fusiform face area (FFA) activity
might be associated with a rise in V1 activity. Accordingly, effectively learning their dynamics
patterns requires both capturing cross-voxel and within-voxel time series information. The vanilla
MLP-MIXER [82] can be used to bind information across both of these dimensions, but the human
brain has unique traits that make directly applying vanilla MLP-MIXER insufficient/impractical.
First, there does not exist in general a canonical grid of the brain to encode voxel activities, which
makes patch extraction challenging. Second, contrary to images that can be divided into patches
of the same size, the partitioning of voxels might not be all the same size due to the complex brain
topology. Third, vanilla MLP-MIXER employs a fixed static mixing matrix for binding patches,
while in the brain the functionality of each patch is important and a different set of patchs should be
mixed differently based on their connections and functionality. To address these challenges, the *VA
Encoder* employs two submodules, *time-mixer* and *voxel-mixer* with dynamic mixing matrix, to fuse
information across both time and voxel dimensions, respectively.

The human brain is comprised of functional systems (FS) [76], which are groups of voxels that
perform similar functions [80]. We take advantage of this hierarchical structure and patch voxels
based on their functionality. However, the main challenge is that the sizes of the patches (set of voxels
with similar functionality) are different. To this end, inspired by the inference of ViT models [28], we
linearly interpolate patches with smaller sizes.

**Functional Patching.** Let $|\mathcal{V}|$ be the number of voxels and $\mathbf{X} \in \mathbb{R}^{|\mathcal{V}| \times (T \times t_{\max})}$ represents the
time series of voxels activities over all time windows. We split $\mathbf{X}$ to spatio-temporal patches $\mathbf{X}_i$
with size $|f_i| \times t_{\max}$, where $f_i \in \mathbb{F}$ is a functional system [76]. To address the challenge of
different patch sizes, we use INTERPOLATE(.) to linearly interpolate patches to the same size $N_p$:
i.e., $\tilde{\mathbf{X}}_i = \text{INTERPOLATE}(\mathbf{X}_i)$, where $\tilde{\mathbf{X}}_i \in \mathbb{R}^{N_p \times t_{\max}}$. We let $\tilde{\mathbf{X}} \in \mathbb{R}^{|\mathcal{V}| \times t_{\max}}$ be the matrix of $\tilde{\mathbf{X}}_i$.

**Voxel-Mixer.** Since the effect of each task (e.g., in task-based fMRI) on brain activity as well as
the time it lasts varies [98], for different tasks, we might need to emphasize more on a subset of
voxels. To this end, to bind information across voxels, we use a dynamic attention mechanism that
uses a learnable dynamic mixing matrix $\mathbf{P}_i$, learning to mix a set of input voxels based on their
functionality. While using different learnable matrices for mixing voxels activity provides a more
powerful architecture, its main challenge is a large number of parameters. To mitigate this challenge,
we first reduce the dimensions of $\tilde{\mathbf{X}}$, split it into a set of segments, denoted as $S$, and then combine
the transformed matrices. Given a segment $s \in S$ we have:

$$\hat{\mathbf{X}}^{(t)(s)} = \tilde{\mathbf{X}}^{(t)} \, \mathbf{W}^{(s)}_{\text{segment}} \quad \in \mathbb{R}^{|\mathcal{V}| \times d}, \qquad\qquad (\textit{Dimension Reduction})$$

$$\mathbf{P}^{(s)}_i = \text{SOFTMAX}\left( \text{FLAT}\left( \hat{\mathbf{X}}^{(t)(s)} \right) \mathbf{W}^{(s)(i)}_{\text{flat}} \right) \quad \in \mathbb{R}^{1 \times |\mathcal{V}|}, \qquad (\textit{Learning Dynamic Mixer})$$

$$\mathbf{X}^{(t)}_{\text{PE}} = \left[ \big\|_{s \in S} \mathbf{P}^{(s)} \tilde{\mathbf{X}}^{(t)(s)} \right] \mathbf{W}_{\text{PE}} \quad \in \mathbb{R}^{|\mathcal{V}| \times t_{\max}}, \qquad (\textit{Dynamic Positional Encoding})$$

$$\mathbf{H}^{(t)}_{\text{Voxel}} = \text{Norm}\left( \tilde{\mathbf{X}}^{(t)} \right) + \text{SIGMOID}\left( \frac{\mathbf{X}^{(t)}_{\text{PE}} \, \mathbf{X}^{(t)\top}_{\text{PE}}}{\sqrt{\tilde{T}}} \right) \mathbf{X}^{(t)}_{\text{PE}}, \qquad (\textit{Dynamic Self-Attention})$$

where $\mathbf{W}^{(s)}_{\text{segment}} \in \mathbb{R}^{t_{\max} \times d}$, $\mathbf{W}^{(s)(i)}_{\text{flat}} \in \mathbb{R}^{d|\mathcal{V}| \times |\mathcal{V}|}$, $\mathbf{W}_{\text{PE}} \in \mathbb{R}^{t_{\max} \times t_{\max}}$ are learnable parameters, $\|$ is
concatenation, and SIGMOID(.) is row-wise sigmoid normalization. Note that for different segments
we use different dimensionality reduction matrices to reinforce the power of the Voxel Mixing.

**Time Mixer.** We then fuse information in the time dimension by using the Time Mixer submodule.
To this end, the Time Mixer employs a 2-layer MLP with layer-normalization [4]:

$$\mathbf{H}^{(t)}_{\textit{Time}} = \mathbf{H}^{(t)}_{\text{Voxel}} + \left( \sigma \left( \text{LayerNorm}\left( \mathbf{H}^{(t)}_{\text{Voxel}} \right) \mathbf{W}^{(1)}_{\text{Time}} \right) \mathbf{W}^{(2)}_{\text{Time}} \right) \in \mathbb{R}^{|\mathcal{V}| \times t_{\max}}, \qquad (1)$$

where $\mathbf{W}_{\text{Time}}^{(1)}$ and $\mathbf{W}_{\text{Time}}^{(1)}$ are learnable matrices, $\sigma(.)$ is an activation function (we use GeLU [38]), and `LayerNorm` is layer normalization [4].

## 3.2 Functional Connectivity Encoder

To encode the functional connectivity graph, we design an MLP-based architecture that learns both the structural and temporal properties of the graph. Inspired by the recent success of all-MLP architecture in graphs [21], we extend MLP-MIXER to temporal graphs. We first define patches in temporal graphs. While patches in images, videos, and multivariate timeseries can simply be non-overlapping regular grids, patches in graphs are overlapping non-grid structures, which makes the patching extraction challenging. He et al. [35] suggest using graph partitioning algorithms to extract graph patches; however, these partitioning algorithms ① only consider structural properties, missing the temporal dependencies, and ② can be time-consuming, limiting the scalability to dense graphs like brain functional connectome. To this end, we propose a temporal-patch extraction algorithm such that nodes (voxels) in each patch share similar temporal and structural properties.

**Temporal Patching.** To extract temporal patches from the graph, we use a biased temporal random walk that walks over both nodes (voxels) and timestamps. Given a functional connectivity graph $\mathcal{G}_F = \{\mathcal{G}_F^{(t)}\}_{t=1}^T$, we sample $M$ walks with length $m+1$ started from node (voxel) $v_0 \in \mathcal{V}$ like: $\mathcal{W}alk : (v_0, t_0) \rightarrow (v_1, t_1) \rightarrow \cdots \rightarrow (v_m, t_m)$, such that $(v_{i-1}, v_i) \in \mathcal{E}^{(t_i)}$, and $t_0 \geq t_1 \geq t_2 \geq \cdots \geq t_m$. Note that, contrary to some previous temporal random walks [92, 10], we allow the walker to walk in the same timestamp at each step. While backtracking over time, we aim to capture temporal information and extract the dynamics of voxels' activity over *related* timestamps. Previous studies show that doing a task can affect brain activity even after 2 minutes [98]. To this end, since more recent connections can be more informative, we use a biased sampling procedure. Let $v_{\text{pre}}$ be the previously sampled node, we use hyperparameters $\theta, \theta_0 \geq 0$ to sample a node $v$ with probability proportional to $\exp(\theta(t - t_{\text{pre}} + \theta_0))$, where $t$ and $t_{\text{pre}}$ are the timestamps that $(v_{\text{pre}}, v) \in \mathcal{E}^{(t)}$ and the timestamp of the previous sample, respectively. In this sampling procedure, smaller (resp. larger) $\theta$ means less (resp. more) emphasis on recent timestamps. Each walk started from $v$ can be seen as a temporal subgraph, and so we let $\rho_v$ be the union of all these subgraphs (walks started from $v$). We treat each of $\rho_v$ as a temporal patch.

**Temporal Pooling Mixer.** Given the temporal graph patches that we extracted above, we need to encode each patch to obtain patch encodings (we later use these patch encodings as their corresponding voxel's encodings). While simple poolings (e.g., SUM(.)) are shown to miss information [10], more complicated pooling functions consider a static pooling rule. However, as discussed above, the effect of performing a task on the neuroimaging data might last for a period of time and the pooling rule might change over time. To this end, we design a temporal pooling, TPMIXER(.), that dynamically pools a set of voxels in a patch based on their timestamps.

Given a patch $\rho_{v_0} = \{v_0, v_1, \ldots, v_k\}$, for each voxel we consider the correlation of its activity with other voxels' as its preliminary feature vector. That is, for each voxel $v$, we consider its feature vector in the time window $t$ as $\mathcal{A}_v^{(t)}$, the $v$'s corresponding row in $\mathcal{A}^{(t)}$. We abuse the notation and use $\mathcal{A}_{\rho_v}^{(t)}$ to refer to the set of $\mathcal{A}^{(t)}$'s rows corresponding to $\rho_v$. Since patch sizes are different, we zero pad $\mathcal{A}_{\rho_v}^{(t)}$ matrices to a fixed size. Note that this zero padding is important to capture the size of each voxel neighborhood. The voxel with more zero-padded dimensions in its patch has less correlation with others. To capture both cross-feature and cross-voxel dependencies, we can use the same architecture as the Time Mixer and Voxel-Mixer. However, the main drawback of this approach is that a pooling function is expected to be permutation invariant while the Voxel Mixer phase is permutation variant. To address this challenge, we fuse information across features in a non-parametric manner as follows:

$$\mathbf{H}_{\text{F}}^{(t)} = \mathcal{A}_{\rho_v}^{(t)} + \sigma \left( \texttt{Softmax} \left( \texttt{LayerNorm} \left( \mathcal{A}_{\rho_v}^{(t)} \right)^\top \right) \right)^\top \in \mathbb{R}^{|\rho_v| \times d'}, \tag{2}$$

where $\sigma(.)$ is an activation function, `Softmax(.)` is used to normalize across features to bind and fuse feature-wise information in a non-parametric manner, avoiding permutation variant operations, and $d'$ is the feature vector size. To dynamically fuse information across voxels, we use the same idea as

dynamic self-attention in §3.1 and learn dynamic matrices $\mathbf{P}_{\text{Pool}_i}$; let $d_{\text{patch}}$ be the patch size:

$$\mathbf{P}_{\text{Pool}_i} = \text{SOFTMAX}\left(\text{FLAT}\left(\mathbf{H}_{\text{F}}^{(t)}\right)\mathbf{W}_{\text{Pool}}^{(i)}\right) \in \mathbb{R}^{1 \times d'} \tag{3}$$

$$\mathbf{h}_{\rho_v} = \text{MEAN}\left(\text{Norm}(\mathbf{H}_{\text{F}}^{(t)}) + \mathbf{H}_{\text{PE}}^{(t)}\ \text{SOFTMAX}\left(\frac{\mathbf{H}_{\text{PE}}^{(t)\top}\mathbf{H}_{\text{PE}}^{(t)}}{\sqrt{d_{\text{patch}}}}\right)\right) \in \mathbb{R}^{1 \times d'}, \tag{4}$$

where $\mathbf{H}_{\text{PE}}^{(t)} = \mathbf{H}_{\text{F}}^{(t)}\mathbf{P}_{\text{Pool}}$ is the transformation of $\mathbf{H}_{\text{F}}^{(t)}$ by dynamic matrix $\mathbf{P}_{\text{Pool}}$.

**Theorem 1.** TPMIXER *is permutation invariant and a universal approximator of multisets.*

**Time Encoding.** To distinguish different timestamps in the functional connectivity graph, we use a non-learnable time encoding module proposed by Cong et al. [21]. This encoding approach helps reduce the number of parameters, and also it has been shown to be more stable and generalizable [21]. Given hyperparameters $\alpha, \beta$, and $d$, we use feature vector $\boldsymbol{\omega} = \{\alpha^{-i/\beta}\}_{i=0}^{d-1}$ to encode each timestamp $t$ using $\cos(\boldsymbol{\omega}t)$ function. Therefore, we obtain the time encoding as $\boldsymbol{\eta}_t = \cos(\boldsymbol{\omega}t)$.

**Voxel-, Edge-, and Graph-level Encodings.** Depending on the downstream task, we might obtain voxel-, edge-, or graph-level encodings. For each voxel $v \in \mathcal{V}$, we let $\mathcal{E}^{(t)}[\rho_v]$ be the set of connections in the patch of $v$. To obtain the voxel-level encoding of each voxel $v$, $\boldsymbol{\psi}_v$, we use patch encoding and concatenate it with all the weighted mean of timestamp encodings; i.e., $\boldsymbol{\psi}_v^t = \text{MLP}([\mathbf{h}_{\rho_v}\|\mathcal{T}_v])$, where $\mathcal{T}_v = \frac{\sum_{t_0=1}^{t}\mathcal{E}^{(t_0)}[\rho_v]\boldsymbol{\eta}_{t_0}}{\sum_{t_0=1}^{t}\mathcal{E}^{(t_0)}[\rho_v]}$. For a connection $e = (u,v) \in \mathcal{E}^{(t)}$, we obtain its encoding by concatenating its endpoints and its timestamp encodings; i.e., $\boldsymbol{\zeta}_{(u,v)}^{(t)} = \text{MLP}\left([\boldsymbol{\psi}_u^t, \boldsymbol{\psi}_v^t, \boldsymbol{\eta}_t]\right)$. Finally, to obtain the graph level encoding, we use vanilla MLP-MIXER [82] on patch encodings; let $\boldsymbol{\Psi}^{(t)}$ be the matrix whose rows are $\boldsymbol{\psi}_v^{(t)}$:

$$\boldsymbol{\Psi}_{\text{patch}}^{(t)} = \boldsymbol{\Psi}^{(t)} + \mathbf{W}_{\text{patch}}^{(2)}\sigma\left(\mathbf{W}_{\text{patch}}^{(1)}\text{LayerNorm}\left(\boldsymbol{\Psi}^{(t)}\right)\right), \tag{5}$$

$$\text{ENC}(\mathcal{G}_F^{(t)}) = \text{MEAN}\left(\boldsymbol{\Psi}_{\text{patch}}^{(t)} + \sigma\left(\text{LayerNorm}\left(\boldsymbol{\Psi}_{\text{patch}}^{(t)}\right)\mathbf{W}_{\text{channel}}^{(1)}\right)\mathbf{W}_{\text{channel}}^{(2)}\right). \tag{6}$$

Similar to the above, to obtain the brain-level encoding, $\mathbf{Z}_{\mathbf{V}}^{(t)}$, based on voxel acitivity timeseries, we use MLP-MIXER on $\mathbf{H}_{Time}^{(t)}$.

### 3.3 Self-supervised Pre-training

In §3.1 and §3.2 we obtained the encodings of the same contexts, from different perspectives. In this section, inspired by [40, 5], we use the mutual information of these two perspectives from the same context, to learn voxel- and brain-level encodings in a self-supervised manner. To this end, let $\boldsymbol{\Psi}$ be the voxel-level encodings obtained from functional connectome, $\mathbf{Z}_{\mathbf{F}}^{(t)} = \text{ENC}(\mathcal{G}_F^{(t)})$ be the global encoding (brain-level) of the functional connectome, $\mathbf{H}_{\text{Voxel}}^{(t)}$ be the voxel activity encodings from the brain activity timeseries, and $\mathbf{Z}_{\mathbf{V}}^{(t)}$ be the global encoding (brain-level) of the voxel activity timeseries, we aim to maximize $I(\mathbf{Z}_{\mathbf{V}}^{(t)}, \boldsymbol{\psi}_{v,i}^{(t)}) + I(\mathbf{Z}_{\mathbf{F}}^{(t)}, (\mathbf{H}_{\text{Time}}^{(t)})_{v,j})$ for all $v \in \mathcal{V}$ and possible $i, j$. Following previous studies [5], we use Noise-Contrastive Estimation (NCE) [34] and minimize the following loss function:

$$\mathbb{E}_{(\mathbf{Z}_{\mathbf{F}}^{(t)}, \boldsymbol{\psi}_{v,i}^{(t)})}\left[\mathbb{E}_{\mathcal{N}}\left[\mathcal{L}_{\Phi}(\mathbf{Z}_{\mathbf{F}}^{(t)}, \boldsymbol{\psi}_{v,i}^{(t)}, \mathcal{N})\right]\right] + \mathbb{E}_{(\mathbf{Z}_{\mathbf{V}}^{(t)}, (\mathbf{H}_{\text{Voxel}}^{(t)})_{v,j})}\left[\mathbb{E}_{\mathcal{N}}\left[\mathcal{L}_{\Phi}(\mathbf{Z}_{\mathbf{V}}^{(t)}, (\mathbf{H}_{\text{Voxel}}^{(t)})_{v,j}, \mathcal{N})\right]\right], \tag{7}$$

where $\mathcal{N}$ is the set of negative samples, $(\mathbf{Z}_{\mathbf{V}}^{(t)}, \boldsymbol{\psi}_{v,i}^{(t)})$ and $(\mathbf{Z}_{\mathbf{F}}^{(t)}, (\mathbf{H}_{\text{Voxel}}^{(t)})_{v,j})$ are the positive sample pairs, and $\mathcal{L}_{\Phi}$ is a standard Log-Softmax.

## 4 Experiments

**Dataset.** We use six real-world datasets: ① We present BVFC, a task-based fMRI dataset that includes voxel activity timeseries and functional connectivity of 3 subjects when looking at the 8460

Table 1: Performance on multi-class brain classification: Mean ACC (%) ± standard deviation.

| Methods | Bvfc | Bvfc-MEG | HCP-Mental | HCP-Age |
|---|---|---|---|---|
| USAD | $48.52_{\pm1.94}$ | $50.02_{\pm1.13}$ | $73.49_{\pm1.56}$ | $39.17_{\pm1.68}$ |
| HyperSAGCN | $51.92_{\pm1.47}$ | $51.19_{\pm1.88}$ | $90.37_{\pm1.61}$ | $47.38_{\pm1.96}$ |
| GMM | $53.11_{\pm1.44}$ | $53.04_{\pm1.73}$ | $90.92_{\pm1.83}$ | $47.75_{\pm1.26}$ |
| GraphMixer | $53.17_{\pm1.21}$ | $53.12_{\pm1.18}$ | $91.13_{\pm1.44}$ | $48.32_{\pm1.11}$ |
| BrainNetCnn | $49.10_{\pm1.83}$ | $50.12_{\pm1.57}$ | $83.58_{\pm1.68}$ | $42.26_{\pm2.03}$ |
| BrainGNN | $50.63_{\pm1.67}$ | $51.08_{\pm0.96}$ | $85.25_{\pm2.17}$ | $43.08_{\pm1.54}$ |
| FbNetGen | $50.18_{\pm0.98}$ | $50.94_{\pm1.39}$ | $84.47_{\pm1.88}$ | $42.83_{\pm1.78}$ |
| Admire | $54.36_{\pm1.39}$ | $54.87_{\pm1.92}$ | $89.74_{\pm1.93}$ | $47.82_{\pm1.72}$ |
| PTGB | $55.89_{\pm1.78}$ | $55.11_{\pm1.62}$ | $92.58_{\pm1.31}$ | $48.41_{\pm1.47}$ |
| BNTransformer | $55.03_{\pm1.35}$ | $55.17_{\pm1.74}$ | $91.71_{\pm1.48}$ | $47.94_{\pm1.15}$ |
| BrainMixer | $\mathbf{67.24_{\pm1.47}}$ | $\mathbf{62.58_{\pm1.12}}$ | $\mathbf{96.32_{\pm0.29}}$ | $\mathbf{57.83_{\pm1.03}}$ |

images from 720 categories. This data is based on THINGS dataset [37]. ② Bvfc-MEG is the MEG counterpart of Bvfc. ③ ADHD [63] contains data for 250 subjects in the ADHD group and 450 subjects in the typically developed (TD) control group. ④ The Seizure detection TUH-EEG dataset [78] consists of EEG data (31 channels) of 642 subjects. ⑤ ASD [23] contains data for 45 subjects in the ASD group and 45 subjects in the TD group. ⑥ HCP [87] contains data from 7440 neuroimaging samples each of which is associated with one of the seven ground-truth mental states.

**Evaluation Tasks.** In our experiments we focus on 4 downstream tasks: ① Edge-Anomaly Detection (AD), ② Voxel AD, ③ Brain AD, and ④ Brain Classification. For the edge and voxel AD tasks, we follow previous studies [8, 59], and inject 1% and 5% anomalous edges into the functional connectivity in the control group. For brain AD all datasets has ground-truth anomalies (see Appendix E.2). The ground truth anomalies in Bvfc are the brain responses to not recognizable images, generated by BigGAN [12], and for other datasets are brain activity of people living with ADHD, seizure, and ASD. For brain classification, we focus on the prediction of ① categories of images seen by the subjects (in Bvfc, and Bvfc-MEG), and ② age prediction and mental state decoding (in HCP-Age, and HCP-Mental). The details of the setup are in Appendix E.

**Baselines.** We compare BrainMixer with state-of-the-art time series, graph, and brain anomaly detection and learning models: ① Graph-based methods: GOutlier [1], NetWalk [100], Hyper-SAGCN [104], Graph MLP-Mixer (GMM) [35], GraphMixer [21]. ② brain-network-based methods: BrainGnn [56], FbNetGen [46], BrainNetCnn [49], ADMire [9], and BNTransformer [47], PTGB [99]. ③ Time-series-based methods: USAD [2], Time Series Transformer (TST) [103], and Mvts [69]. We may exclude some baselines in some tasks as they cannot be applied in that setting. We use the same training procedure as BrainMixer. The details are in Appendix E.1.

**Brain Classification.** Table 1 reports the performance of BrainMixer and baselines on multi-class brain classification tasks. BrainMixer achieves the best accuracy on all datasets with 14.3% average improvement (20.3% best improvement) over the best baseline. There are three main reasons for BrainMixer's superior performance: ① While the time series-based model only uses voxel activity timeseries, and graph-based methods only use functional connectivity graph, BrainMixer takes advantage of both and learns the brain representation at different levels of granularity, which can provide complementary information. ② Static methods (e.g., PTGB, BrainGnn, etc.), miss the dynamics of brain activity, while BrainMixer employs a time encoding module to learn temporal properties. ③ Compared to graph learning methods (e.g., GMM, GraphMixer, etc.), BrainMixer is specifically designed for the brain, taking advantage of its special properties.

**Anomaly Detection.** Table 2 reports the performance of BrainMixer and baselines on anomaly detection tasks at different scales: i.e., edge-, voxel-, and brain-level. BrainMixer achieves the best AUC-PR on all datasets with 6.2%, 5.7%, 4.81% average improvement over the best baseline in edge AD, voxel AD, and brain AD, respectively. The main reasons for this superior performance are as above. Note that brain-level anomaly detection can also be seen as a brain classification task. However, here, based on the nature of the data, we separate these two tasks.

**How Does BrainMixer Detect GAN Generated Images?** The visual cortex, responsible for processing visual information, is hierarchically organized with multiple layers building upon simpler features at lower stages [86]. Initially, neurons detect edges and colors, but on deeper levels, they specialize in recognizing more complex patterns and objects. Figure 2 (Left) (resp. (Right)) reports the distribution of detected voxel activity by BrainMixer when the subject looking at non-recognizable images (resp. natural images). Interestingly, while the distributions share similar patterns in lower

Table 2: Performance on anomaly detection: Mean AUC-PR (%) $\pm$ standard deviation[†].

| | Methods | BVFC | BVFC-MEG | HCP 1% | HCP 5% | ADHD 1% | ADHD 5% | TUH-EEG 1% | TUH-EEG 5% | ASD 1% | ASD 5% |
|---|---|---|---|---|---|---|---|---|---|---|---|
| Edge-level AD | GOUTLIER | $65.12_{\pm2.97}$ | $59.45_{\pm2.61}$ | $62.47_{\pm1.15}$ | $61.83_{\pm1.28}$ | $65.37_{\pm0.93}$ | $64.70_{\pm2.09}$ | $65.61_{\pm1.82}$ | $64.12_{\pm0.97}$ | $60.85_{\pm0.97}$ | $59.13_{\pm1.86}$ |
| | NETWALK | $71.67_{\pm1.56}$ | $62.75_{\pm1.16}$ | $73.12_{\pm1.25}$ | $72.19_{\pm1.31}$ | $70.29_{\pm2.15}$ | $69.86_{\pm2.58}$ | $71.14_{\pm1.36}$ | $70.27_{\pm1.42}$ | $69.07_{\pm2.20}$ | $68.52_{\pm2.55}$ |
| | HYPERSAGCN | $80.17_{\pm1.59}$ | $70.83_{\pm1.27}$ | $82.94_{\pm1.14}$ | $81.98_{\pm1.58}$ | $84.22_{\pm1.61}$ | $83.96_{\pm1.47}$ | $73.99_{\pm0.83}$ | $72.65_{\pm0.97}$ | $73.26_{\pm1.08}$ | $73.18_{\pm0.92}$ |
| | GRAPHMIXER | $87.13_{\pm0.99}$ | $75.91_{\pm1.59}$ | $86.87_{\pm1.96}$ | $86.19_{\pm1.48}$ | $85.12_{\pm1.46}$ | $84.86_{\pm1.58}$ | $75.93_{\pm0.95}$ | $75.12_{\pm1.08}$ | $84.91_{\pm2.27}$ | $83.52_{\pm2.03}$ |
| | BRAINNETCNN | $80.92_{\pm1.18}$ | $71.54_{\pm2.07}$ | $80.79_{\pm1.23}$ | $79.44_{\pm1.18}$ | $80.58_{\pm1.62}$ | $79.95_{\pm2.01}$ | $73.06_{\pm1.74}$ | $72.87_{\pm1.31}$ | $72.68_{\pm2.12}$ | $72.01_{\pm1.45}$ |
| | BRAINGNN | $81.96_{\pm1.76}$ | $72.68_{\pm1.13}$ | $82.15_{\pm1.84}$ | $81.38_{\pm1.61}$ | $79.02_{\pm1.85}$ | $78.64_{\pm1.43}$ | $72.96_{\pm1.58}$ | $71.73_{\pm1.14}$ | $72.14_{\pm1.25}$ | $71.82_{\pm1.73}$ |
| | FBNETGEN | $81.58_{\pm1.92}$ | $72.66_{\pm1.52}$ | $82.05_{\pm1.19}$ | $81.53_{\pm1.82}$ | $79.89_{\pm1.63}$ | $78.97_{\pm1.84}$ | $73.04_{\pm1.53}$ | $72.56_{\pm1.33}$ | $72.51_{\pm1.28}$ | $71.62_{\pm1.82}$ |
| | ADMIRE | $87.12_{\pm1.61}$ | $75.91_{\pm1.43}$ | $87.01_{\pm1.27}$ | $86.38_{\pm1.17}$ | $86.23_{\pm1.74}$ | $85.18_{\pm2.21}$ | $76.68_{\pm1.82}$ | $75.14_{\pm1.67}$ | $86.52_{\pm1.72}$ | $85.44_{\pm1.49}$ |
| | PTGB | $86.52_{\pm1.64}$ | $75.93_{\pm1.71}$ | $86.83_{\pm1.59}$ | $86.00_{\pm1.28}$ | $86.14_{\pm1.15}$ | $85.22_{\pm1.21}$ | $75.98_{\pm1.16}$ | $74.92_{\pm1.08}$ | $86.18_{\pm1.58}$ | $85.72_{\pm1.05}$ |
| | BNTRANSFORMER | $86.61_{\pm1.72}$ | $75.82_{\pm1.18}$ | $86.22_{\pm1.77}$ | $85.15_{\pm1.12}$ | $85.83_{\pm1.97}$ | $85.14_{\pm1.67}$ | $75.91_{\pm1.72}$ | $75.24_{\pm1.53}$ | $74.92_{\pm1.18}$ | $74.11_{\pm1.37}$ |
| | BRAINMIXER | $\mathbf{91.62_{\pm1.36}}$ | $\mathbf{82.58_{\pm1.92}}$ | $\mathbf{90.14_{\pm1.72}}$ | $\mathbf{90.02_{\pm1.49}}$ | $\mathbf{91.74_{\pm0.93}}$ | $\mathbf{91.48_{\pm1.41}}$ | $\mathbf{80.91_{\pm1.19}}$ | $\mathbf{80.85_{\pm1.62}}$ | $\mathbf{90.44_{\pm1.57}}$ | $\mathbf{90.27_{\pm1.39}}$ |
| Voxel-level AD | USAD | $68.27_{\pm1.16}$ | $62.73_{\pm1.27}$ | $65.49_{\pm1.31}$ | $65.01_{\pm1.18}$ | $72.79_{\pm1.48}$ | $72.19_{\pm0.94}$ | $72.81_{\pm1.42}$ | $71.36_{\pm1.03}$ | $66.28_{\pm1.16}$ | $65.17_{\pm1.15}$ |
| | TST | $70.62_{\pm1.48}$ | $68.57_{\pm1.81}$ | $69.18_{\pm1.64}$ | $69.11_{\pm1.32}$ | $74.81_{\pm1.14}$ | $73.99_{\pm1.47}$ | $73.71_{\pm1.55}$ | $73.03_{\pm1.47}$ | $69.23_{\pm1.82}$ | $68.94_{\pm1.73}$ |
| | MVTS | N/A | N/A | N/A | N/A | N/A | N/A | $77.48_{\pm1.81}$ | $77.02_{\pm1.29}$ | N/A | N/A |
| | GOUTLIER | $64.66_{\pm2.38}$ | $60.17_{\pm1.25}$ | $63.59_{\pm1.62}$ | $63.07_{\pm1.52}$ | $68.97_{\pm1.16}$ | $67.12_{\pm0.93}$ | $65.18_{\pm1.09}$ | $65.01_{\pm1.57}$ | $59.67_{\pm1.42}$ | $58.49_{\pm1.35}$ |
| | NETWALK | $68.73_{\pm1.16}$ | $63.61_{\pm1.31}$ | $66.98_{\pm1.44}$ | $66.04_{\pm1.63}$ | $75.16_{\pm1.23}$ | $74.73_{\pm1.01}$ | $72.21_{\pm0.91}$ | $71.62_{\pm1.46}$ | $71.28_{\pm1.17}$ | $71.02_{\pm1.49}$ |
| | HYPERSAGCN | $78.84_{\pm1.22}$ | $71.62_{\pm1.96}$ | $80.74_{\pm1.51}$ | $79.18_{\pm1.83}$ | $83.94_{\pm1.13}$ | $83.01_{\pm0.92}$ | $75.62_{\pm1.12}$ | $74.83_{\pm0.78}$ | $74.93_{\pm1.47}$ | $74.15_{\pm1.19}$ |
| | GRAPHMIXER | $76.94_{\pm1.68}$ | $71.44_{\pm1.39}$ | $81.55_{\pm1.82}$ | $81.07_{\pm1.27}$ | $81.37_{\pm1.09}$ | $80.83_{\pm1.16}$ | $72.95_{\pm1.26}$ | $72.01_{\pm0.82}$ | $72.49_{\pm1.28}$ | $72.27_{\pm1.69}$ |
| | BRAINNETCNN | $80.17_{\pm1.49}$ | $73.91_{\pm1.54}$ | $82.75_{\pm1.27}$ | $82.21_{\pm1.73}$ | $82.79_{\pm1.08}$ | $81.12_{\pm1.16}$ | $73.98_{\pm1.24}$ | $73.01_{\pm1.08}$ | $73.18_{\pm0.95}$ | $72.88_{\pm1.04}$ |
| | BRAINGNN | $79.92_{\pm1.63}$ | $73.25_{\pm1.94}$ | $82.99_{\pm1.65}$ | $82.13_{\pm1.66}$ | $81.14_{\pm1.05}$ | $80.83_{\pm0.87}$ | $73.06_{\pm1.14}$ | $72.74_{\pm0.86}$ | $72.54_{\pm1.38}$ | $71.12_{\pm1.19}$ |
| | FBNETGEN | $79.17_{\pm2.04}$ | $72.35_{\pm1.84}$ | $82.26_{\pm1.37}$ | $81.62_{\pm1.49}$ | $80.91_{\pm1.12}$ | $80.94_{\pm1.74}$ | $72.53_{\pm1.48}$ | $72.06_{\pm1.29}$ | $72.11_{\pm1.94}$ | $71.28_{\pm1.22}$ |
| | PTGB | $85.18_{\pm1.83}$ | $76.16_{\pm1.08}$ | $85.72_{\pm1.14}$ | $84.95_{\pm1.33}$ | $86.43_{\pm1.16}$ | $86.36_{\pm1.15}$ | $77.54_{\pm1.37}$ | $77.32_{\pm1.21}$ | $77.92_{\pm1.26}$ | $77.76_{\pm1.25}$ |
| | BNTRANSFORMER | $85.19_{\pm1.23}$ | $75.67_{\pm1.14}$ | $85.02_{\pm0.96}$ | $84.36_{\pm1.59}$ | $86.13_{\pm1.21}$ | $86.11_{\pm1.82}$ | $77.96_{\pm1.32}$ | $77.08_{\pm1.06}$ | $76.05_{\pm1.52}$ | $75.72_{\pm1.18}$ |
| | BRAINMIXER | $\mathbf{90.14_{\pm1.57}}$ | $\mathbf{81.52_{\pm1.32}}$ | $\mathbf{89.27_{\pm1.61}}$ | $\mathbf{88.94_{\pm1.24}}$ | $\mathbf{89.97_{\pm1.14}}$ | $\mathbf{89.81_{\pm1.27}}$ | $\mathbf{79.45_{\pm1.19}}$ | $\mathbf{79.23_{\pm0.94}}$ | $\mathbf{89.51_{\pm1.78}}$ | $\mathbf{89.24_{\pm1.59}}$ |
| Brain-level AD | USAD | $71.93_{\pm1.15}$ | $61.32_{\pm1.71}$ | $67.79_{\pm2.28}$ | $67.36_{\pm2.61}$ | $82.87_{\pm2.03}$ | $80.52_{\pm1.84}$ | $72.03_{\pm1.17}$ | $71.48_{\pm1.05}$ | $71.62_{\pm1.58}$ | $70.98_{\pm1.41}$ |
| | TST | $72.47_{\pm1.23}$ | $67.12_{\pm2.07}$ | $67.94_{\pm1.69}$ | $67.22_{\pm1.17}$ | $83.54_{\pm1.38}$ | $83.04_{\pm1.12}$ | $72.96_{\pm1.39}$ | $72.11_{\pm1.58}$ | $72.76_{\pm1.71}$ | $72.04_{\pm1.56}$ |
| | MVTS | N/A | N/A | N/A | N/A | N/A | N/A | $83.53_{\pm1.91}$ | $82.41_{\pm1.02}$ | N/A | N/A |
| | NETWALK | $72.16_{\pm1.44}$ | $69.57_{\pm1.73}$ | $69.14_{\pm1.49}$ | $68.66_{\pm1.52}$ | $83.11_{\pm1.02}$ | $82.81_{\pm1.61}$ | $71.06_{\pm1.05}$ | $69.94_{\pm1.12}$ | $72.85_{\pm1.17}$ | $72.21_{\pm1.34}$ |
| | HYPERSAGCN | $80.25_{\pm1.15}$ | $76.91_{\pm1.18}$ | $72.26_{\pm1.47}$ | $72.01_{\pm1.21}$ | $86.94_{\pm1.63}$ | $86.17_{\pm1.49}$ | $75.31_{\pm0.85}$ | $74.79_{\pm1.09}$ | $76.72_{\pm1.32}$ | $75.81_{\pm1.58}$ |
| | GMM | $81.79_{\pm1.24}$ | $77.84_{\pm1.52}$ | $74.87_{\pm1.58}$ | $74.02_{\pm1.10}$ | $85.89_{\pm0.98}$ | $85.03_{\pm1.18}$ | $76.62_{\pm1.17}$ | $76.11_{\pm1.26}$ | $76.37_{\pm1.83}$ | $75.68_{\pm1.59}$ |
| | GRAPHMIXER | $82.56_{\pm1.19}$ | $77.91_{\pm1.26}$ | $75.03_{\pm1.72}$ | $74.46_{\pm1.53}$ | $86.02_{\pm1.15}$ | $85.64_{\pm1.09}$ | $77.49_{\pm1.09}$ | $76.63_{\pm1.22}$ | $76.82_{\pm1.84}$ | $76.18_{\pm1.80}$ |
| | BRAINNETCNN | $78.47_{\pm1.18}$ | $73.12_{\pm1.27}$ | $70.73_{\pm1.77}$ | $70.12_{\pm1.86}$ | $85.84_{\pm0.96}$ | $85.07_{\pm1.52}$ | $73.92_{\pm0.97}$ | $73.07_{\pm1.51}$ | $75.96_{\pm1.66}$ | $75.03_{\pm1.28}$ |
| | BRAINGNN | $79.81_{\pm1.57}$ | $75.28_{\pm1.61}$ | $72.98_{\pm1.55}$ | $72.41_{\pm1.16}$ | $84.59_{\pm1.26}$ | $83.72_{\pm1.35}$ | $72.41_{\pm1.38}$ | $71.55_{\pm1.16}$ | $75.12_{\pm1.33}$ | $74.57_{\pm1.52}$ |
| | FBNETGEN | $78.94_{\pm1.24}$ | $74.49_{\pm1.33}$ | $71.62_{\pm1.53}$ | $71.06_{\pm1.48}$ | $84.67_{\pm1.26}$ | $84.08_{\pm1.37}$ | $72.69_{\pm1.18}$ | $71.87_{\pm1.12}$ | $75.34_{\pm1.21}$ | $74.73_{\pm1.39}$ |
| | ADMIRE | $83.72_{\pm1.18}$ | $78.83_{\pm1.56}$ | $75.52_{\pm1.81}$ | $74.59_{\pm1.12}$ | $86.27_{\pm1.72}$ | $85.18_{\pm1.56}$ | $78.12_{\pm1.47}$ | $77.59_{\pm1.68}$ | $77.18_{\pm1.61}$ | $76.33_{\pm1.45}$ |
| | PTGB | $84.08_{\pm1.35}$ | $79.68_{\pm1.62}$ | $76.01_{\pm1.07}$ | $75.13_{\pm1.48}$ | $87.59_{\pm1.12}$ | $86.99_{\pm0.96}$ | $79.17_{\pm1.36}$ | $78.64_{\pm1.55}$ | $80.56_{\pm1.29}$ | $80.04_{\pm1.16}$ |
| | BNTRANSFORMER | $83.86_{\pm1.52}$ | $79.03_{\pm1.78}$ | $75.64_{\pm1.82}$ | $75.09_{\pm1.18}$ | $87.54_{\pm1.04}$ | $86.92_{\pm1.48}$ | $79.36_{\pm1.71}$ | $78.08_{\pm1.16}$ | $77.19_{\pm2.01}$ | $76.58_{\pm1.73}$ |
| | BRAINMIXER | $\mathbf{88.13_{\pm1.27}}$ | $\mathbf{84.59_{\pm1.70}}$ | $\mathbf{80.67_{\pm1.13}}$ | $\mathbf{80.49_{\pm1.07}}$ | $\mathbf{91.38_{\pm0.94}}$ | $\mathbf{90.98_{\pm1.02}}$ | $\mathbf{85.74_{\pm1.16}}$ | $\mathbf{85.63_{\pm1.23}}$ | $\mathbf{89.14_{\pm1.54}}$ | $\mathbf{88.99_{\pm1.15}}$ |

[†] We perform statistical comparison with baselines via paired $t$-tests. Shaded blue indicates significance improvement over the baselines ($p$-value $\leq 0.05$), while gray shaded boxes indicate ($p$-value $> 0.05$). The maximum $p$-value is 0.061.

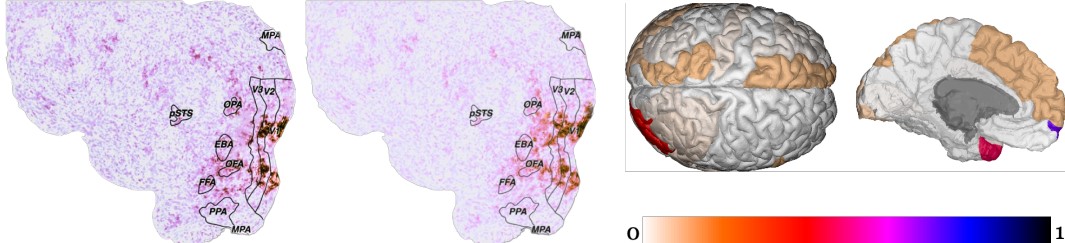

Figure 2: Distribution of abnormal voxel activities detected by BRAINMIXER in the visual cortex when seeing (Left) GAN-generated, (Right) Normal image.

Figure 3: The distribution of detected abnormal voxels by BRAINMIXER in condition ADHD group.

levels (e.g., V1 and V2 voxels), higher-level voxels (e.g., V3) are less active when the subject sees non-recognizable images. These results show the potential of BRAINMIXER in learning meaningful representation of voxels activity. Additional details can be found in Appendix F.

**Case Study: ADHD** We first train our model on the neuroimages of the TD group and test it on the ADHD group to detect abnormal voxel activities that might be correlated to ADHD symptoms. Figure 3 reports the distribution of anomalous voxels within the brain of the ADHD group. 78% of all found abnormal voxel activities by BRAINMIXER are located in the Frontal Pole, Left and Right Temporal Poles, and Lingual Gyrus. Surprisingly, these findings are consistent with previous studies on ADHD, which use diffusion tensor imaging [54] and Forman–Ricci curvature changes [17].

## 5 Conclusion

In this work, we present an unsupervised pre-training framework, BRAINMIXER, that bridges the representation learning of voxel activity and functional connectivity by maximizing their mutual information. The experimental results show the potential of BRAINMIXER in ① detecting abnormal brain activity that might cause a brain disease/disorder, ② disease/disorder detection, and ③ understanding object representation in the brain.

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

# Appendix

## Table of Contents

# A Backgrounds

We begin by reviewing the preliminaries and background concepts that we refer to in the main paper.

## A.1 Graphs and Machine Learning on Graphs

**Temporal Graphs.** We first define the concept of temporal graphs, which are graphs such that each connection is associated with a timestamp. We formally define temporal graph as follows:

**Definition 1** (Temporal Graphs). *Let $\mathcal{G} = (\mathcal{V}, \mathcal{E}, \mathcal{T})$ be a temporal graph, where $\mathcal{V}$ is the set of nodes, $\mathcal{T}$ is the set of timestamps, and $\mathcal{E} \subseteq \mathcal{V} \times \mathcal{V} \times \mathcal{T}$ is the set of edges. That is, each connection between two nodes $(u, v)$ is associated with a timestamp like $t \in \mathcal{T}$.*

In this study, we use snapshot based representation of temporal graphs. That is, $\mathcal{G} = \{\mathcal{G}^{(t)}\}$ is the set of graphs, where $\mathcal{G}^{(t)} = (\mathcal{V}, \mathcal{E}^{(t)})$ represents the state of the graph $\mathcal{G}$ at timestamp $t$.

**Node Representation Learning.** Node representation learning in graphs is a process that aim to map nodes of a graph into a vector space. This representation seeks to capture the structure of the graph, the features of the nodes, their dynamic over time, and their relationships. The core idea is to represent each node with a vector that encapsulates not just its own attributes but also its position, dynamics, and role within the larger graph structure.

**Definition 2** (Node Representation Learning). *Let $\mathcal{G} = (\mathcal{V}, \mathcal{E})$ be a graph with nodes $\mathcal{V}$ and edges $\mathcal{E}$. The goal is to learn a function $f : \mathcal{V} \to \mathbb{R}^d$, where $d$ is the dimension of the target vector space (usually $d \ll |V|$, implying a lower-dimensional representation).*

## A.2 Voxel Time Series Activity

Neuroimaging modalities (e.g., fMRI, MEG) provide (estimated) recordings of neural activity signals. To this end, their estimation is built up in a 3-D image building block, units called voxels, which represent a small cube of brain tissue. For example, fMRI measures the blood-oxygen level dependence (BOLD) of each voxel in order to estimate the neural activity of the whole brain over time. In the literature, for each voxel, most studies aggregate its activity (e.g., its BOLD) over each time window, called beta weight [73, 88, 72]. However, this approach misses the voxel activity dynamic over each task. In this study, we consider voxels' activity as it is (without aggregation) and model it as timeseries data. We model this data as a multivariate time seris: An fMRI scan involves thousands of voxels, leading to a multivariate time series $\{X_1(t), X_2(t), ..., X_n(t)\}$ where $X_i(t)$ is the time series for the $i$-th voxel.

## A.3 Brain Functional Connectivity

The brain's functional connectivity is a graph, derived from a neuroimaging modality (often fMRI), where each node represents a brain parcel or ROI, and two nodes are connected if there is a statistical association between their functionality. In more details, as discussed above, fMRI measures brain activity by detecting changes in blood flow. The primary data from fMRI is the Blood Oxygen Level Dependent (BOLD) signal, reflecting changes in the oxygenation level of the blood. Deriving a brain network from fMRI data involves ① preprocessing, ② parcellation using atlases, and ③ computing correlations.

1. Preprocessing: Is the sequence of actions to clean the data and make it for process. Common preprocessing techniques are:

   - **Motion Correction:** Aligns all the neuroimages to a reference neuroimage to correct for patient movement.
   - **Band-Pass Filtering:** Isolating the frequency band that corresponds to the fMRI signal (usually 0.01 to 0.1 Hz).
   - **Slice Timing Correction:** Adjusts for the time difference in image acquisition between slices.
   - **Smoothing:** Applies a Gaussian filter to reduce spatial noise and improve signal-to-noise ratio.

2. Parcellation using Atlases: Brain atlases divide the brain into regions of interest (ROIs). Each ROI represents a node in the brain network. Common atlases include:
   - **AAL (Automated Anatomical Labeling):** Divides the brain into areas based on anatomical structures.
   - **Harvard-Oxford Atlas:** Based on probabilistic information from a large population.
   - **Functional Atlases:** Based on functional connectivity patterns, e.g., resting-state networks.
3. Computing Correlations: For each ROI defined by the atlas, the time series of the fMRI signal is extracted. This step involves averaging the fMRI signal over all voxels within each ROI (Note that in this paper, we argue that the best case is to consider each voxel as an ROI). Next, to capture the statistical association of ROIs activity, we compute the Pearson correlation coefficient between the time series of every pair of ROIs:

$$\mathbf{C}_{ij} = \frac{\sum_{k=1}^{T}(X_i(k) - \bar{X}_i)(X_j(k) - \bar{X}_j)}{\sqrt{\left(\sum_{k=1}^{T}(X_i(k) - \bar{X}_i)^2\right)\left(\sum_{k=1}^{T}(X_j(k) - \bar{X}_j)^2\right)}}, \tag{8}$$

Here, $C_{ij}$ is the correlation between ROI $i$ and ROI $j$, and $X_i$, $X_j$ are the time series for ROIs $i$ and $j$, respectively. To construct a network $\mathcal{G} = (\mathcal{V}, \mathcal{E})$, where $\mathcal{V}$ is the set of ROI and $\mathcal{E}$ is the set of connections, a threshold is applied to the correlation matrix. Only correlations above a certain value are considered to represent connections in $\mathcal{E}$.

In EEG and MEG data the process is the same while each signal corresponds to a channel and so in the constructed brain network, each node is a channel and each connection shows high Pearson's correlation between its endpoints.

### A.4 MLP-MIXER

The MLP-MIXER architecture [82] is a novel neural network design that has attracted much attention in the field of computer vision. It presents itself as a distinctive alternative to the CNNs and Transformer models. The structure of MLP-MIXER is composed of two key sub-layers in each layer: the patch mixing layer and the channel mixing layer. The patch mixing layer processes spatial information within each channel independently, whereas the channel mixing layer combines the information across various channels. This dual process of mixing is crucial for the MLP-MIXER's capability to detect both local and global image dependencies.

The mathematical representation of the MLP-MIXER is as follows:

patch Mixer:

$$\mathbf{H}_{\text{patch}} = \mathbf{E} + \mathbf{W}_{\text{patch}}^{(2)}\sigma\left(\mathbf{W}_{\text{patch}}^{(1)}\texttt{LayerNorm}(\mathbf{E})^{\top}\right)^{\top}, \tag{9}$$

Channel Mixer:

$$\mathbf{H}_{\text{channel}} = \mathbf{H}_{\text{patch}} + \mathbf{W}_{\text{channel}}^{(2)}\sigma\left(\mathbf{W}_{\text{channel}}^{(1)}\texttt{LayerNorm}(\mathbf{H}_{\text{patch}})\right), \tag{10}$$

**Challenges of Extending MLP-MIXER to Graphs and Time Series.** In graphs, the vanilla MLP-MIXER [82] can be used to bind information across both of feature and node dimensions, but directly applying vanilla MLP-MIXER to graphs is insufficient and impractical. First, there does not exist in general a canonical grid of the graphs (contrary to images) to encode nodes, which makes patch extraction challenging. Second, contrary to images that can be divided into patches of the same size, the partitioning of nodes in graphs might not be all the same size due to the complex graph topology. Moreover, in temporal graphs, dynamics of the graph and its temporal properties should be captured to effectively learn its node encodings. The vanilla MLP-MIXER is not capable of learning temporal dynamics.

Similarly, in multivariate time series data, there is no canonical grid and patches are not necessarily the same size.

**Challenges of using MLP-MIXER and its variants for Brain Activity.** While there are existing studies that aim to address the above limitations and define patches in graphs using graph partitioning

Table 3: The differences of VA Encoder and FC Encoder with MLP-MIXER.

| | MLP-MIXER [82] | FC Encoder | VA Encoder |
|---|---|---|---|
| Patches |  |  |  |
| Input | Images (2-d Regular grid) Same width and height | Graphs (Irregular and non-grid) Different sizes (#Nodes, #Edges) | Time series (Unstructured) Variable Length |
| Patch Extraction | Based on pixels order Non-overlapping patches Same patches at each epoch | Based on Temporal Random Walks Overlapping patches Different patches at each epoch | Based on Brain Functional Systems Overlapping patches Different patches at each epoch |
| Patch Encoder | Same patch size Usign 2-d MLP | Variable patch size Using TPMIXER (1-d MLP + Dynamic self-attention) | Variable patch size Using Voxel-Mixer (2-d MLP + Dynamic self-attention) |
| Positional Information | Implicitly ordered | No universal ordering Permutation invariant | No universal ordering Permutation variant |

algorithms [35], or first-hop neighbourhood [21], there are designed for general cases and miss special properties of the brain. ① The human brain is comprised of functional systems [76], which are groups of voxels that perform similar functions [80]. Recently, Trockman & Kolter [83] show that the main power of vision architectures like MLP-MIXER and ViT [50] comes from patching, splitting the image into multiple same size parts which might show the same concept. Inspired by this, we suggest using functional patching in analysis of brain activity, i.e. splitting voxels into some groups such that each group has similar functionality. ② This approach results in another challenge which is the size of functional systems are not the same and simply using vanilla MLP-MIXER on functional patches is not feasible. To this end, we present an interpolation method to linearly interpolate functional systems to the same size.

The main differences of our VA Encoder and FC Encoder with MLP-MIXER are summarized in Table 3. In this table, 1-d MLP and 2-d MLP refers to applying MLP in one of the dimensions and both dimensions, respectively.

## B  BVFC Dataset

In this section, we introduce BVFC dataset and discuss how it is derived from the fMRI and MEG data.

### B.1  THINGS Dataset

The Things dataset[2] [37] is a large battery of visual object recognition datasets that use a common set of images as visual stimuli. These datasets cover a broad range of data types, ranging from behavioral aspects, such as similarity judgments, to neural responses to the presented images, such as fMRI and MEG recordings. The shared image database includes more than 26,000 images in total, categorized into 1,854 object concepts. In this work, we used ① the THINGS fMRI1 dataset, consisting of event-related BOLD responses of three human subjects to 8,460 images selected from 720 categories (12 images per each). ② the THINGS MEG1 dataset, consisting of Magnetoencephalography (MEG) of 4 subjects for 22,248 images (1,854 categories, 12 images per category), collected over the course of 12 sessions. Both preprocessed and raw versions of the fMRI and MEG datasets are provided by

---

[2]https://things-initiative.org

the Things authors[3]. However, in the preprocessed version of fMRI dataset, each voxel is associated with a single beta value, which misses the dynamic of voxel activity over time. For the purpose of this work, we utilized the raw version as we required the time series of fMRI recordings and follow the following preprocess steps:

## B.2 Preprocessing

The beta values provided in the preprocessed version of the THINGS fMRI dataset are single measures of each voxel's response to a certain stimulus, which are obtained by applying a general linear model (GLM) to the voxels' time series. Since the preprocessed dataset only offers beta values, we utilized and preprocessed the raw data without applying the GLM step at the end. The preprocessing pipeline used by the authors of THINGS also includes a semi-supervised ICA-denoising step, which requires prior experience with fMRI noise-signal classification. We replaced this stage with ICA-AROMA [70], an automatic ICA-denoising tool, to improve the replicability of our results without the need for manual supervision or intervention in the denoising step. For the rest of the preprocessing steps, we followed the same pipeline used by Hebart et al. [37]. For each image, we use 13 seconds of fMRI signals of the human subject, and treat each as a time window. We use the output of the preprocess time series as the voxel level brain activity. To derive the brain functional connectivity from the time series data, we consider each voxel as an ROI and calculate the statistical association of the time series of two voxels $v_i$ and $v_j$ in each time window using Pearson's correlation:

$$\mathbf{C}_{ij} = \frac{\sum_{k=1}^{T}(X_i(k) - \bar{X}_i)(X_j(k) - \bar{X}_j)}{\sqrt{\left(\sum_{k=1}^{T}(X_i(k) - \bar{X}_i)^2\right)\left(\sum_{k=1}^{T}(X_j(k) - \bar{X}_j)^2\right)}}, \tag{11}$$

where $X_i(k)$ and $X_j(k)$ are $v_i$ and $v_j$ activities at time $k$, and $\bar{X}_i$ and $\bar{X}_j$ are their average activity over the time window, respectively. We next, for each voxel removes negative elements and then keep 90-percentile of its corresponding correlation. We use the same approach on the time series of channels in MEG to obtain brain connectivity networks. For the preprocessing scripts visit this link.

## B.3 Image Classification

Understanding object representation in the brain is a key step toward revealing the basic building blocks of human visual processing [37]. Toward this direction, in the first downstream task on BVFC we aim to classify seen images during the fMRI and MEG recording. As discussed above, the fMRI dataset consists of responses of three human subjects to 8,460 images selected from 720 categories (12 images per each) from THINGS database [36]. Each of the images has a high-level concept as its high-level label, which described the type of the object in the image (e.g., "Food", "Human Body", etc.)[4]. In the first task, we aim to predict the high-level label of the seen image by using the fMRI responses of the human subject. This task is a multi-class classification tasks with 9 classes.

## B.4 Anomaly Detection

In the fMRI1 THINGS dataset [37], there are 100 unique catch images that were created using the generative adversarial neural network, BigGAN [12]. These images were generated by interpolating between two latent vectors, yielding novel objects that were not recognizable. We take advantage of these images and design a downstream task to detect these images.

The visual cortex, responsible for processing visual information, is hierarchically organized with multiple layers building upon simpler features at lower stages [86]. Initially, neurons detect edges and colors, but on deeper levels, they specialize in recognizing more complex patterns and objects. Accordingly, we expect our model to detect GAN generated images by using the subject's brain fMRI response. We model this task as a binary classification task, where the brain fMRI response to

---

[3]https://plus.figshare.com/collections/THINGS-data_A_multimodal_collection_of_large-scale_datasets_for_investigating_object_representations_in_brain_and_behavior/6161151

[4]Note that the high-level labels of images are different from their original labels, as each high-level class include a set of primary labels. For example, all "Pizza", "Fast Food", and "Bacon" are in a high-level class of "Food".

each natural image is considered "normal" and the brain fMRI response to GAN generated images is considered "Abnormal". For further information about the generated images by GAN and its architecture see the original paper of THINGS dataset [37] and original paper of BigGAN [12].

## C Additional Related Work

To further situate our BRAINMIXER in a broader context, we briefly review self-supervised representation learning of brain activity and time series representation learning.

**Self-supervised Representation Learning of Brain Activity.** In representation learning of brain activity, such as fMRI, MEG and EEG, obtaining labeled data is challenging and costly. To address this, various self-supervised learning techniques have been introduced. Banville et al. [6] suggest using relative positioning, temporal shuffling, and contrastive predictive coding specifically to EEG data. Additionally, Mohsenvand et al. [65] and Kostas et al. [51] presents an approach to learn EEG signals using negative sampling and contrastive learning, which is only suitable for SEEG and EEG data. Yang et al. [99] propose an unsupervised pre-training technique designed specifically for brain networks using contrastive learning and maximizing the mutual information between an anchor point of investigation $X$ from a data distribution $H$ and its positive samples, while minimizing its mutual information with its negative samples. All these methods are either ① are designed for a specific type of neuroimaging data (e.g., EEG) and cannot be generalized to other neuroimage modalities, ② are based on negative sampling generation which bias the performance toward the patterns of generated negative samples, being unable to generalize to complex and unseen patterns, and/or ③ uses either time series data or connectivity network, missing information from different level of granularity.

**Time Series Representation Learning.** Representation learning of multivariate time series has been getting increasingly popular with the motivation that modeling the complex relationships between covariates should improve the forecasting performance[19]. to this end, Transformers [89] attract much attention due to their superior performance in modeling long-term sequential data. [105] present Informer and [95] present Autoformer to address the efficiency challenges in long-term forecasting. Zhou et al. [107] design FEDformer and later FiLM [106] that decompose the sequences using Fast Fourier Transformation for better extraction of long-term information. Recently, Chen et al. [19] design TSMIXER, and all MLP architecture for time series forecasting. Not only the purpose of these methods are different from VA Encoder, but also their architectures are different from VA Encoder from a subset of following aspects: ① They bind information across time series, missing the cross-time dependencies of signals. ② These methods use fixed static learnable matrices for binding time series, while in the brain signals, the functionality of each time series is important and a different set of signals should be mixed differently based on their corresponding voxel's (channel's) connections and functionality. ③ They treat each time series the same, while in multivariate time series some signals cab be more important than others for a specific downstream task.

**MLP-MIXER for fMRI Data.** MLP-MIXER shows promising performance on image data. One approach to learn from fMRI data is to treat fMRI image in each time window as an image and then employ an MLP-MIXER to learn representation for voxels. Geenjaar et al. [31] designed a fully-differentiable non-linear framework for whole-brain dynamic factor learning and applied MLP-MIXER to fMRI data. However, this study suffers from all the MLP-MIXER limitations that we discussed. For more explanations and illustrative examples see our discussion on the difference of our encoders with MLP-MIXER in Appendix A.4.

### C.1 Our Contributions

As we discussed in Sections 1, 2, and Appendix C, existing studies miss a subset of the following: ① the functional connectivity between voxels, ② timeseries activity of voxels, ③ special properties of the brain like hierarchical structure and its complex dynamics. Here, we summarize our contributions as follows:

1. Voxel Activity Encoder: We present VA Encoder, a novel multivariate time series encoder that employs a dynamic attention mechanism to bind information across both voxel and time dimensions. VA Encoder by learning the representation of each voxel allows us to obtain brain activity encodings at different level of granularity (e.g., voxel- , functional

system-, and/or brain-level encodings). Our experiments (row 5 in Table 6) show that VA Encoder alone, i.e. without using functional connectivity, outperforms baselines in different downstream tasks.

2. Simple and Low Cost, but Effective Patching: We propose functional patching for learning brain activity. While existing patching methods are either ① grid-based and inapplicable to graphs and time series, ② requires additional computational cost, and/or ③ cannot use specific brain properties (e.g., functional systems), missing the functional similarity of voxels. Our functional patching uses additional knowledge about the brain functional systems and patch the brain into some groups in which voxels have similar functionality. Our experimental results show that removing functional patching and replacing it with either random patching or clustering patching can damage the performance (See Appendix F.3 and Table 6).

3. Functional Connectivity Encoder: To encode the functional connectivity graph, we design an MLP-based architecture that learns both the structural and temporal properties of the graph using temporal random walks. FC Encoder first extracts temporal patches using temporal random walks and then fuses information within each patch using a novel dynamic self-attention mechanism. To obtain the brain activity encoding at different level of granularity, we further propose an adaptive permutation invariant pooling method that theoretically is the universal approximator of any multi-set function. Our experimental results in Table 6 show that FC Encoder alone, i.e. without using time series of voxel activity, outperforms baselines in different downstream tasks.

4. Self-Supervised Pre-training Framework: We present a novel self-supervised pre-training framework without using contrastive learning, which requires generating negative samples. Existing pre-trianing methods for the representation learning of brain activity suffers from two main limitations: ① They require negative samples to learn from data in a contrastive manner [99]. However, brain activity is complex by its nature, and simple negative samples cause missing complex patterns, damaging the performance. ② They are based on a meta knowledge about a specific brain disease and so cannot generalize to other neuroimage modalities and different neuroimaging tasks [101]. Our framework allows self-supervised pre-training of any neuroimaging data that provides multivariate time series (e.g., fMRI, EEG, MEG, iEEG, etc.) without using any meta knowledge about the disease or downstream tasks, making it generalizable to different neuroimage modalities and different downstream tasks.

# D   Theoretical Guarantee of TSETMIXER Performance

**Theorem 1.** TPMIXER *is permutation invariant and a universal approximator of multisets.*

*Proof.* Let $\pi(S)$ be an arbitrary permutation of set $S$, we aim to show that $\Psi(S) = \Psi(\pi(S))$. We first recall the TSETMIXER and its two phases: Let $S = \{\mathbf{v}_1, \ldots, \mathbf{v}_d\}$, where $\mathbf{v}_i \in \mathbb{R}^{d_1}$, be the input set and $\mathbf{V} = [\mathbf{v}_1, \ldots, \mathbf{v}_d]^T \in \mathbb{R}^{d \times d_1}$ be its matrix representation: we first fuse information across features in a non-parametric manner as follows:

$$\mathbf{H}_{\mathrm{F}}^{(t)} = \mathbf{V} + \sigma\left(\texttt{Softmax}\left(\texttt{LayerNorm}\left(\mathbf{V}\right)^{\top}\right)\right)^{\top}, \tag{12}$$

Now, for $\pi(S)$, let $\pi(\mathbf{V}) = [\pi(\mathbf{v}_1), \ldots, \pi(\mathbf{v}_d)]^T \in \mathbb{R}^{d \times d_1}$ be its matrix representation, for the first phase we have:

$$\pi(\mathbf{V}) + \sigma\left(\texttt{Softmax}\left(\texttt{LayerNorm}\left(\pi(\mathbf{V})\right)^\top\right)\right)^\top \tag{13}$$

$$= \pi(\mathbf{V}) + \sigma\left(\texttt{Softmax}\left(\pi\left(\texttt{LayerNorm}\left(\mathbf{V}\right)^\top\right)\right)\right)^\top \tag{14}$$

$$= \pi(\mathbf{V}) + \pi\left(\sigma\left(\texttt{Softmax}\left(\texttt{LayerNorm}\left(\mathbf{V}\right)^\top\right)\right)^\top\right) \tag{15}$$

$$= \pi\left(\mathbf{V} + \sigma\left(\texttt{Softmax}\left(\texttt{LayerNorm}\left(\mathbf{V}\right)^\top\right)\right)^\top\right) \tag{16}$$

$$= \pi\left(\mathbf{H}_{\text{F}}^{(t)}\right), \tag{17}$$

which means that the first phase of TSETMIXER is equivariant. In the above, we used the fact that $\texttt{Softmax}$ is permutation equivariant. In the second part, we first have:

$$\mathbf{P}_{\text{Pool}_i} = \text{SOFTMAX}\left(\text{FLAT}\left(\mathbf{H}_{\text{F}}^{(t)}\right)\mathbf{W}_{\text{Pool}}^{(i)}\right) \tag{18}$$

$$\Rightarrow \quad \text{SOFTMAX}\left(\text{FLAT}\left(\pi(\mathbf{H}_{\text{F}}^{(t)})\right)\mathbf{W}_{\text{Pool}}^{(i)}\right) \tag{19}$$

$$= \text{SOFTMAX}\left(\pi\left(\text{FLAT}\left(\mathbf{H}_{\text{F}}^{(t)}\right)\right)\mathbf{W}_{\text{Pool}}^{(i)}\right) \tag{20}$$

$$= \pi\left(\text{SOFTMAX}\left(\text{FLAT}\left(\mathbf{H}_{\text{F}}^{(t)}\right)\mathbf{W}_{\text{Pool}}^{(i)}\right)\right) \tag{21}$$

$$= \pi\left(\mathbf{P}_{\text{Pool}_i}\right). \tag{22}$$

Also, note that we defined $\mathbf{H}_{\text{PE}}^{(t)}$ as $\mathbf{H}_{\text{PE}}^{(t)} = \mathbf{H}_{\text{F}}^{(t)}\mathbf{P}_{\text{Pool}}$. Therefore, we have:

$$\text{MEAN}\left(\text{Norm}\left(\pi\left(\mathbf{H}_{\text{F}}^{(t)}\right)\right) + \pi\left(\mathbf{H}_{\text{PE}}^{(t)}\right) \ \text{SOFTMAX}\left(\pi\left(\frac{\mathbf{H}_{\text{PE}}^{(t)^\top}\mathbf{H}_{\text{PE}}^{(t)}}{\sqrt{d_{\text{patch}}}}\right)\right)\right) \tag{23}$$

$$= \text{MEAN}\left(\pi\left(\text{Norm}\left(\mathbf{H}_{\text{F}}^{(t)}\right) + \mathbf{H}_{\text{PE}}^{(t)} \ \text{SOFTMAX}\left(\frac{\mathbf{H}_{\text{PE}}^{(t)^\top}\mathbf{H}_{\text{PE}}^{(t)}}{\sqrt{d_{\text{patch}}}}\right)\right)\right) \tag{24}$$

$$= \mathbf{h}_{\rho_v} \tag{25}$$

In the last step, we use the fact that $\text{MEAN}(.)$ is permutation invariant, which results TSETMIXER to be permutation invariant.

Since the patch mixer is just normalization it is inevitable and cannot affect the expressive power of TSETMIXER. Also, channel mixer is a 2-layer MLP with attention, which are the universal approximator of any function. Therefore, due to the fact that TSETMIXER is permutation invariant, we can conclude that it is a universal approximator of multi-set functions. $\qquad\square$

# E  Experimental Setup

## E.1  Baselines

Since BRAINMIXER combines functional connectivity and voxel timeseries activity, we compare our method to fourteen previous state-of-the-art methods and baselines on the timeseries, functional connectivity, and graph encoding:

1. GOutlier [1]: GOutlier uses a generative model for edges in a node cluster and labels outliers as anomaly.

2. NETWALK [100]: Yu et al. [100] proposed a random-walk based dynamic graph embedding approach, NETWALK. NETWALK first uses simple random walks and jointly minimizes the pairwise distance of vertex representations of each sampled walk. Next, it uses a clustering-based technique to dynamically detect network anomalies.

3. HYPERSAGCN [104]: HyperSAGCN (Self-attention-based graph convolutional network for hypergraphs) utilizes a spectral aggregated graph convolutional network to refine the embeddings of nodes within each hyperedge. HyperSAGCN generates initial node embeddings by hypergraph random walks and combines node embeddings by MEAN(.) pooling to compute the embedding of hyperedge. The model with code can be found in here.

4. Graph MLP-Mixer (GMM) [35]: Graph MLP-Mixer uses graph partitioning algorithms to split the input graph into overlapping graph patches (subgraphs) and then employs a graph neural network to encode each patch. It then uses an MLP to fuse information across patch encodings. The model with code can be found in here. Note that Graph MLP-Mixer cannot take advantage of temporal properties of the graph as it is designed for static networks.

5. GRAPHMIXER [21]: GRAPHMIXER is a simple method that concatenates the 1-hop temporal connections and their time encoding of each node as its representative matrix. It then uses an MLP-MIXER to encode each representative matrix to obtain node encodings. The model with code can be found in here.

6. FBNETGEN [46]: FBNETGEN is a graph neural network based generative model for functional brain networks from fMRI data that includes three components: a dimension reduction phase to denoise the raw time-series data, a graph generator for brain networks generation from the encoded features, and a GNN predictor for predictions based on the generated brain networks. The model with code can be found in here.

7. BRAINGNN [56]: BRAINGNN is a graph neural network-based framework that maps regional and cross-regional functional connectivity patterns. Li et al. [56] propose a novel clustering-based embedding method in the graph convolutional layer as well as a graph node pooling to learn ROI encodings in the brain. The model with code can be found in here.

8. BRAINNETCNN [49]: BRAINNETCNN is a CNN-based approach that uses novel edge-to-edge, edge-to-node and node-to-graph convolutional filters that leverage the topological locality of structural brain networks.

9. ADMIRE [9]: ADMIRE is a random walk-based approach that models brain connectivity networks as multiplex graphs. It uses inter-view and intra-view walks to capture the causality between different neuroimage modalities or different frequency band filters.

10. BNTRANSFORMER [47]: BNTRANSFORMER adapts Transformers [89] to brain networks, so it can use unique properties of brain networks. BNTRANSFORMER use connection profiles as node features to provide low-cost positional information and then learns pair-wise connection strengths among ROIs with efficient attention weights. It further uses a novel READOUT operation based on self-supervised soft clustering and orthonormal projection. The model with code can be found in here.

11. PTGB [99]: PTGB is an unsupervised pre-training method designed specifically for brain networks using contrastive learning and maximizing the mutual information between an anchor point of investigation $X$ from a data distribution $H$ and its positive samples, while minimizing its mutual information with its negative samples. The model with code can be found in here.

12. USAD [2]: USAD is an unsupervised representation learning method in time series, which utilizes an encoder-decoder architecture within an adversarial training framework that allows it to take advantage of both.

13. Time Series Transformer (TST) [103]: TST is a transformer-based framework for unsupervised representation learning of multivariate time series, which is capable of pre-training and can be employed on varius downstream tasks.

14. MVTS [69]: MVTS is an unsupervised transformer-based model for time series learning, which utilizes special properties of EEGs for seizure identification. It uses an autoencoder mechanism involving a transformer encoder and an unsupervised loss function for training.

We use the same hyperparameter selection process as BRAINMIXER. Also, we fine tune their training parameters as their original papers using grid search. For the sake of fair comparison, we use the same training, testing and validation data for all the baselines (including same data augmentation and negative sampling). Also, for PTGB [99], which also is capable of pre-training, we use the same datasets and settings as we use for BRAINMIXER.

Table 4: Datasets statistics.

| Datasets | Number of Graphs | Average Number of Nodes | Average Number of Edges | Number of Classes (Multi-class Classification) | Ground-Truth Anomaly (Binary Classification) |
|---|---|---|---|---|---|
| BVFC | 25380 | 11776 | 352479 | 9 | Yes |
| BVFC-MEG | 88992 | 272 | 9841 | 9 | Yes |
| ADHD | 700 | 400 | 6194 | - | Yes |
| TUH-EEG | 642 | 31 | 252 | - | Yes |
| ASD | 90 | 400 | 5903 | - | Yes |
| HCP | 7440 1067 | 1000 | 7635 8041 | 7 (Mental states) 5 (Age) | Yes |

## E.2 Datasets

We use six real-world datasets with different neuroimage modalities and downstream tasks, whose descriptions are as follows:

- BVFC (This Paper): The main characteristics and pre-processing steps are mentioned in Appendix B. For the multi-class classification task, we aim to predict the label of the seen image (9 labels) using the fMRI response of a human subject (3 subjects). For the edge- and node-level anomaly detection tasks, we use synthetic injected anomalies and for the graph anomaly detection, we aim to detect GAN generated images using the fMRI response. We label brain activities that correspond to seeing a GAN generated image (resp. natural image) as "Anomalous" (resp. "Normal"). In the experiments, for the sake of efficiency, we remove irrelevant voxels.

- BVFC-MEG (This Paper): The main characteristics and pre-processing steps are mentioned in Appendix B. For the multi-class classification task, we aim to predict the label of the seen image (9 labels) using the MEG response of a human subject (4 subjects). For the edge- and node-level anomaly detection tasks, we use synthetic injected anomalies and for the graph anomaly detection, we aim to detect natural scenes using the MEG response. We label MEG response that correspond to seeing natural scenes as "Anomalous" and seeing other objects as "Normal".

- ADHD [63]: ADHD [63] contains resting-state fMRI of 250 subjects in the ADHD group and 450 subjects in the typically developed (TD) control group. We follow the standard pre-processing steps [25] to obtain brain networks. For the edge and node anomaly detection tasks, we use synthetic anomalies, while for the graph anomaly detection task we label brain networks of the typically developed control group as "Normal" and brain networks of the ADHD group as "Anomalous".

- TUH-EEG [78]: The seizure detection TUH-EEG dataset [78] consists of EEG data with 31 channels of 642 subjects. For the edge and node anomaly detection tasks, we use synthetic anomalies, while for the graph anomaly detection task we label brain networks of people with and without seizure as "Anomalous" and "Normal", respectively.

- ASD [23]: This dataset includes the resting fMRI data taken from the Autism Brain Imaging Data Exchange (ABIDE) [23]; it contains data for 45 subjects (22 female, age $= 25.4 \pm 8.9$ yrs) in the typically developed control group and 45 subjects (23 female, age $= 23.1 \pm 8.1$ yrs) in the ASD group. We have followed the five pre-processing strategies denoted as DPARSF, followed by Band-Pass Filtering. For the edge and node anomaly detection tasks, we use synthetic anomalies, while for the graph anomaly detection task we label brain networks of the typically developed control group as "Normal" and brain networks of the ASD group as "Anomalous".

- HCP [87]: HCP [87] contains data from 7440 neuroimaging samples each of which is associated with one of the seven ground-truth mental states. Following previous studies [74], we define two downstream multi-class classification tasks: ① Mental states prediction, in which we aim to predict the mental state using the fMRI. ② We aim to predict the age of human subjects using their fMRI. In this tasks, we split the age into 5 groups, balancing the number of samples in each class. Similar to other datasets, we use synthetic anomalies for the edge and node anomaly detection tasks.

Table 5: Hyperparameters used in the grid search.

| Datasets | Sampling Number $M$ | Sampling Time Bias $\theta$ | Temporal Walk Length $m$ | Hidden dimensions |
|---|---|---|---|---|
| BVFC | 4, 8, 16, 32, 64, 128 | $\{0.5, 2.0, 20, 200, 2000, 20000\} \times 10^{-7}$ | 1, 2, 3, 4 | 32, 64, 128 |
| BVFC-MEG | 4, 8, 16, 32, 64 | $\{0.5, 2.0, 20, 200, 2000, 20000\} \times 10^{-7}$ | 1, 2, 3, 4, 5, 6 | 32, 64, 128 |
| ADHD | 4, 8, 16, 32, 64, 128 | $\{0.5, 2.0, 20, 200, 2000, 20000\} \times 10^{-7}$ | 1, 2, 3, 4 | 32, 64, 128 |
| TUH-EEG | 4, 8, 16, 32, 64 | $\{0.5, 2.0, 20, 200, 2000, 20000\} \times 10^{-7}$ | 1, 2, 3, 4, 5, 6 | 32, 64, 128 |
| ASD | 4, 8, 16, 32, 64 | $\{0.5, 2.0, 20, 200, 2000, 20000\} \times 10^{-7}$ | 1, 2, 3, 4 | 32, 64, 128 |
| HCP | 4, 8, 16, 32, 64 | $\{0.5, 2.0, 20, 200, 2000, 20000\} \times 10^{-7}$ | 1, 2, 3, 4 | 32, 64, 128 |

We model the pre-processed fMRI, MEG, and EEG signals as multivariate time series and use them as our time series activity. Next, we discuss how we derive the brain connectivity network.

**Constructing Brain Connectivity Network.** To construct the brain connectivity networks for each dataset, we use the same process as we do for BVFC. We use the output of the preprocess time series as the voxel level (channel level for EEG and MEG) brain activity. To derive the brain connectivity from the time series data, we consider each voxel (or channel) as an ROI and calculate the statistical association of the time series of two voxels (channels) $v_i$ and $v_j$ in each time window using Pearson's correlation:

$$\mathbf{C}_{ij} = \frac{\sum_{k=1}^{T}(X_i(k) - \bar{X}_i)(X_j(k) - \bar{X}_j)}{\sqrt{\left(\sum_{k=1}^{T}(X_i(k) - \bar{X}_i)^2\right)\left(\sum_{k=1}^{T}(X_j(k) - \bar{X}_j)^2\right)}}, \tag{26}$$

where $X_i(k)$ and $X_j(k)$ are $v_i$ and $v_j$ activities at time $k$, and $\bar{X}_i$ and $\bar{X}_j$ are their average activity over the time window, respectively. We next, for each voxel (channel) removes negative elements and then keep 90-percentile of its corresponding correlation.

### E.3 Implementation, Training, and Hyperparameters Tuning

In each task, we split the data into training set (70% of the data), validation set (10% of the data), and test set (20% of the data). In the pre-training phase, we use both training set and validation set to train the model and then we fine tune the pre-trained model for downstream tasks using only training set. For downstream tasks, we use validation set to tune hyperparameters as discussed bellow. During the training of both pre-trained model and fine tuning for downstream tasks, the test set is untouched and it is used only for the final evaluation of the method.

In addition to hyperparameters and modules (activation functions) mentioned in the main paper, here, we report the training hyperparameters of BRAINMIXER: On all datasets, we use a batch size of 32 and use a learning rate of $10^{-3}$. We use the maximum training epoch number of 100 with an early stopping strategy to stop training if the validation performance does not increase for more than 7 epochs. Furthermore, a dropout layers with rate = 0.1 is employed in all neural networks. To tune the model's hyperparameters, we systematically perform grid search. The search domains of each hyperparameter are reported in Table 5.

BRAINMIXER is implemented by `PyTorch` in `Python` and a Linux machine with GPU and 16GB of RAM is used to run evaluations.

Note that we use the same training pipeline as BRAINMIXER for all the baselines. For the sake of fair comparison, we use the same training, testing and validation data for all the baselines (including same data augmentation and negative sampling). Also, for PTGB [99], which also is capable of pre-training, we use the same datasets and settings as we use for pre-training of BRAINMIXER.

### E.4 Data Augmentation & Negative Samples

MLP-MIXER-based architectures are known to have the potenial of overfitting [58]. To mitigate this, we perform data augmentation. For $\mathcal{G}_F^{(t)} = (\mathcal{V}, \mathcal{E}^{(t)})$, in patch extraction, we randomly mask $p$ connections and then we sample temporal walks to generate new patches. Note that, at the end, each patch is an induced subgraph and might include masked connections as well. Furthermore, to generate negative samples: ① To corrupt the functional connectivity, we randomly change one

Table 6: Ablation study on BRAINMIXER. AUC-PR scores on edge AD and ACC on classification.

| Methods | BVFC | | BVFC-MEG | | HCP | | ADHD | |
|---|---|---|---|---|---|---|---|---|
| | Edge AD | Classification | Edge AD | Classification | Edge AD | Classification | Edge AD | Classification |
| BRAINMIXER | $91.62_{\pm1.36}$ | $67.24_{\pm1.47}$ | $82.58_{\pm1.92}$ | $62.68_{\pm1.12}$ | $90.02_{\pm1.49}$ | $96.32_{\pm0.29}$ | $91.48_{\pm1.41}$ | $90.98_{\pm1.02}$ |
| Without Pre-training | $88.75_{\pm2.16}$ | $63.58_{\pm2.09}$ | $80.21_{\pm1.63}$ | $61.02_{\pm1.37}$ | $88.14_{\pm1.29}$ | $93.81_{\pm0.92}$ | $90.18_{\pm1.13}$ | $89.27_{\pm1.06}$ |
| Without VA Encoder | $87.99_{\pm2.04}$ | $59.14_{\pm4.51}$ | $78.52_{\pm2.18}$ | $60.53_{\pm1.83}$ | $86.97_{\pm2.05}$ | $92.41_{\pm1.24}$ | $88.29_{\pm1.41}$ | $88.76_{\pm1.19}$ |
| Replace VA Encoder with TST | $89.02_{\pm1.18}$ | $62.89_{\pm1.49}$ | $80.46_{\pm2.00}$ | $61.78_{\pm1.24}$ | $89.01_{\pm0.86}$ | $93.53_{\pm1.78}$ | $90.06_{\pm1.55}$ | $88.94_{\pm1.98}$ |
| Without FC Encoder | $84.27_{\pm4.37}$ | $65.82_{\pm2.18}$ | $77.09_{\pm3.41}$ | $59.73_{\pm1.12}$ | $85.59_{\pm2.47}$ | $91.64_{\pm1.58}$ | $86.97_{\pm1.16}$ | $87.62_{\pm2.16}$ |
| Replace FC Encoder with BNTRANSFORMER | $87.18_{\pm2.03}$ | $66.12_{\pm1.27}$ | $78.85_{\pm1.36}$ | $62.01_{\pm0.87}$ | $86.76_{\pm1.44}$ | $94.24_{\pm1.25}$ | $88.03_{\pm1.24}$ | $88.81_{\pm0.98}$ |
| Without Functional Patching | $86.35_{\pm2.97}$ | $60.42_{\pm3.53}$ | $77.21_{\pm1.93}$ | $60.28_{\pm1.72}$ | $86.14_{\pm3.09}$ | $91.97_{\pm1.88}$ | $87.51_{\pm1.86}$ | $88.25_{\pm2.53}$ |
| Replace Functional Patching with Partitioning | $88.56_{\pm1.42}$ | $66.50_{\pm1.92}$ | $79.26_{\pm1.51}$ | $60.63_{\pm1.87}$ | $87.55_{\pm1.29}$ | $96.14_{\pm1.04}$ | $90.10_{\pm1.78}$ | $89.69_{\pm1.46}$ |
| Replace TPMIXER by MEAN(.) | $88.51_{\pm1.03}$ | $63.38_{\pm1.48}$ | $78.94_{\pm1.85}$ | $60.91_{\pm2.01}$ | $87.52_{\pm1.91}$ | $93.31_{\pm1.73}$ | $89.04_{\pm0.95}$ | $89.11_{\pm1.52}$ |
| Static Self-Attention | $88.39_{\pm1.40}$ | $63.01_{\pm2.10}$ | $78.63_{\pm1.97}$ | $60.78_{\pm1.64}$ | $87.04_{\pm1.53}$ | $92.95_{\pm1.49}$ | $88.96_{\pm1.22}$ | $88.83_{\pm2.07}$ |
| Remove Time Encoding | $89.58_{\pm0.81}$ | $66.14_{\pm1.52}$ | $79.91_{\pm1.75}$ | $61.19_{\pm1.36}$ | $88.82_{\pm2.07}$ | $94.12_{\pm1.92}$ | $90.57_{\pm0.91}$ | $89.99_{\pm1.04}$ |
| fix $\theta = 0$ | $83.60_{\pm4.52}$ | $59.33_{\pm2.58}$ | $75.96_{\pm2.05}$ | $59.11_{\pm1.46}$ | $85.39_{\pm1.52}$ | $90.51_{\pm1.38}$ | $86.24_{\pm2.01}$ | $87.18_{\pm1.94}$ |

endpoint of a subset of connections. ② To corrupt the timeseries, we follow existing studies [102, 94] on timeseries and replace a brain signal in time window $t$ with another signal that is randomly selected from the batch. Given a pre-trained model $\mathcal{M}$, for different downstream tasks in a semi-supervised setting, we fine-tune $\mathcal{M}$ using a small subset of labeled training data. Also, for each voxel, we concatenate its encodings from VA and FC Encoders.

### E.5 Injecting Synthetic Anomalies

Due to the nature of anomaly detection tasks, specifically in neuroimaging data, there is a lack of unified definition for abnormal brain activity and so the ground truth labels are not available. To mitigate this challenge in evaluation of our approach, we first evaluate our approach using synthetic anomalies and then we perform case studies on real-world data and show that the found anomalies are compatible with previous findings.

① Injecting Abnormal Connections to Brain Connectivity Network: We randomly choose 5% (or 1%, depends on the setting) normal connections and corrupt them. Given a connection $(u, v)$, we randomly change one of its endpoint, assume that $u$, to another voxel like $w$ such that $v$ and $w$ have not been connected previously. Therefore, from a normal connection $(u, v)$, we generate an abnormal connection $(w, v)$. This method is used to evaluate the performance of anomaly detection methods in different domains [59], including neuroimaging [8].

② Injecting Abnormal Activity to Brain Activity Time Series: We randomly choose 5% (or 1%, depends on the setting) of time series and corrupt them as follows: Given a voxel activity time series $X_i$ and $X_j$, during time window $t$, we swap this part of these two randomly selected signals and construct $\tilde{X}_i$ and $\tilde{X}_j$ which are corrupted of $X_i$ and $X_j$, respectively.

Note that only edge-AD and node-AD tasks are evaluated using synthetic data and graph-level anomaly detection methods are evaluated using ground-truth anomalies.

### E.6 Visualization Tools

To visualize the average distribution of anomalous connections, we use BrainPainter [61] with the Desikan-Killiany atlas. Also, to visualize the average distribution of brain activities in the visual cortex we use Pycortex, which is an interactive surface visualizer for fMRI [30].

## F Additional Experimental Results

### F.1 Ablation Study

We next conduct ablation studies on the BVFC, BVFC-MEG, HCP, and ADHD datasets to validate the effectiveness of BRAINMIXER's critical components. Table 6 shows AUC-PR for edge AD and accuracy for classification tasks. The first row reports the performance of the complete BRAINMIXER implementation with pre-training. Each subsequent row shows results for BRAINMIXER with one module modification: row 2 removes the pre-training phase, row 3 removes the VA Encoder module, row 4 replace VA Encoder with TST, row 5 removes FC Encoder module, row 6 replace it with BNTRANSFORMER, row 7 replaces functional patching with random patching, row 8 replace temporal patching with partitioning [48], row 9 replaces TPMIXER with MEAN(.) pooling, row 10

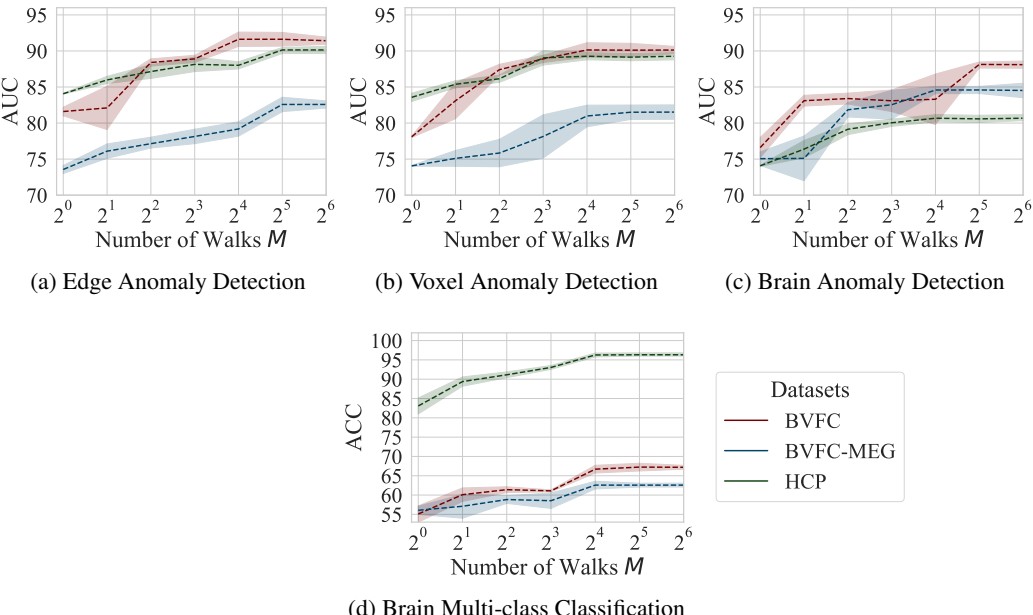

(a) Edge Anomaly Detection     (b) Voxel Anomaly Detection     (c) Brain Anomaly Detection

(d) Brain Multi-class Classification

Figure 4: The effect of the number of walks on the performance of BRAINMIXER in different downstream tasks.

replaces dynamic with static self-attention, row 11 removes time encoder, the last row set $\theta = 0$, removing biased in the sampling. These results show that each component is critical for achieving BRAINMIXER's superior performance. The greatest contribution comes from biased sampling, VA and FC encoders, functional patching, and dynamic self-attention.

### F.2 Parameter Sensitivity

In this section, we evaluate the effect of hyperparameters on the performance of BRAINMIXER in different downstream tasks.

**The Effect of the Number of Walks.** In the first evaluation, we evaluate the effect of the number of walks on the performance of BRAINMIXER. Results are reported in Figure 4. These results show that increasing the number of walks results in better performance. The main reason is that we use the union of walks to capture the neighbourhood of each node over time. The more number of walks the better representation of the temporal neighborhood we can obtain. That is, sampling more walks lets the model to extract more information about the dynamic of nodes neighborhood as well as its structure. Also, notably, we observe that only a small number of sampled walks are needed to achieve competitive performance: in the best case 4 and in the worst case 16 sampled walks are needed to achieve better performance than baselines.

**The Effect of the Walk Length.** In this experiment, we evaluate the effect of the walk length on the performance of BRAINMIXER. Results are reported in Figure 5. The results suggest that the effect of the walk length on performance peaks at a certain point, but the exact value varies with datasets. The main reason for this is that we use walks to capture the structural and temporal properties of each node. Therefore, for dense brain connectivity networks as well as datasets with a large number of *relevant* time windows we need longer walks so the model can extract enough information about both *relevant* time windows and dense neighborhoods. Accordingly, we see increasing trend in BVFC-MEG's performance when we increase the length of the walk. Also, note that increasing the walk length for more sparser brain connectivity networks or for datasets with a smaller number of *relevant* time windows can damage the performance. The reason is we might consider irrelevant time windows by backtracking over time or consider far nodes (voxels in brain connectivity graph), which

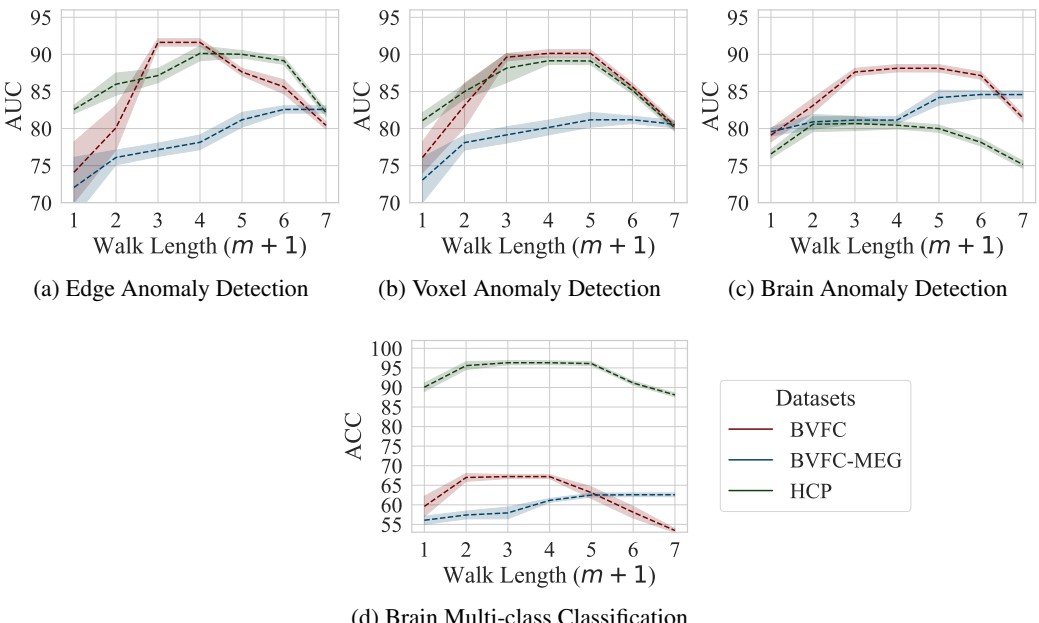

(a) Edge Anomaly Detection  (b) Voxel Anomaly Detection  (c) Brain Anomaly Detection

(d) Brain Multi-class Classification

Figure 5: The effect of the walk length on the performance of BRAINMIXER in different downstream tasks.

are irrelevant. Accordingly, depends on the structure of the brain connectivity graph and temporal properties of time series signals, we might need longer or shorter walks.

**The Effect of the Time Decay $\theta$.** As we discussed in section 3, previous studies show that doing a task can affect brain activity even after 2 minutes [98]. To this end, since more recent connections can be more informative, we use a biased sampling procedure and control the bias using a variable $\theta$. That is, in the proposed sampling procedure, smaller (resp. larger) $\theta$ means less (resp. more) emphasis on recent timestamps.

In this experiment, we evaluate the effect of time decay $\theta$ on the performance of BRAINMIXER. Results are presented in Figure 6. These results suggest that $\theta$ has a dataset-dependent optimal interval. That is, a small $\theta$ means an almost uniform sampling of brain activity history, which results in poor performance when the brain activities in recent time-windows are more important in the dataset. Also, very large $\theta$ might damage the performance as it makes the model focus on the most recent brain activity or only its own time window, missing long-term and lasting brain activities.

Please note that while the value of $\theta$ needs to be tuned to achieve the best performance, choosing arbitrary $\theta$ in a wide proper interval can still results in state-of-the-art performance over baselines.

**The Effect of the Number of ROIs.** The human brain is hierarchically organized and comprised of hierarchical groups of voxels that have similar functionality [80]. Accordingly, different downstream tasks requires studying the brain at different level of granularity. In this experiment, we evaluate the the effect of the number of ROIs[5] on the performance of BRAINMIXER. We vary the number of ROIs from 10000 (voxel-level activity) to 100 (functional system-level activity) and report the results in Figure 7. The results suggest that using more ROIs and so studying the brain at lower-levels like voxel-level can result in a better performance. While there is a little improvement for downstream tasks that are correlated with human brain functional systems (e.g., HCP dataset and classification mental states), the significant improvements are for tasks that are highly correlated to a specific brain region (e.g., BVFC dataset and classifying seen images, which is closely related to human brain visual cortex). As an example, there is a $\approx 50\%$ performance loss in the accuracy of BRAINMIXER on

---

[5]Note that here ROI means any region of interest in the brain not necessarily pre-defined brain regions based on the atlases.

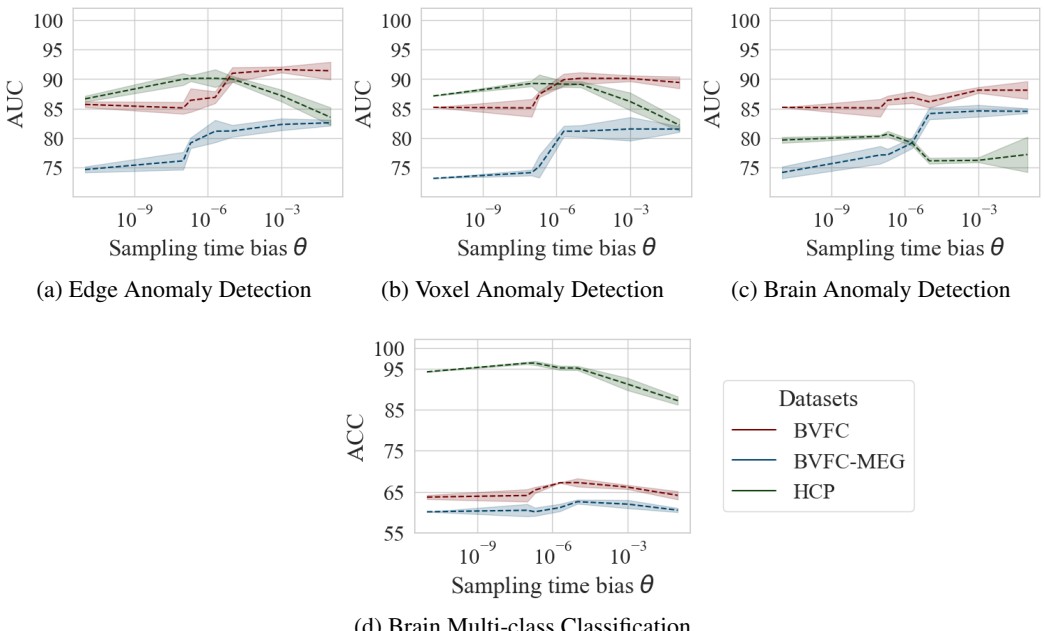

Figure 6: The effect of the time decay $\theta$ on the performance of BRAINMIXER in different downstream tasks.

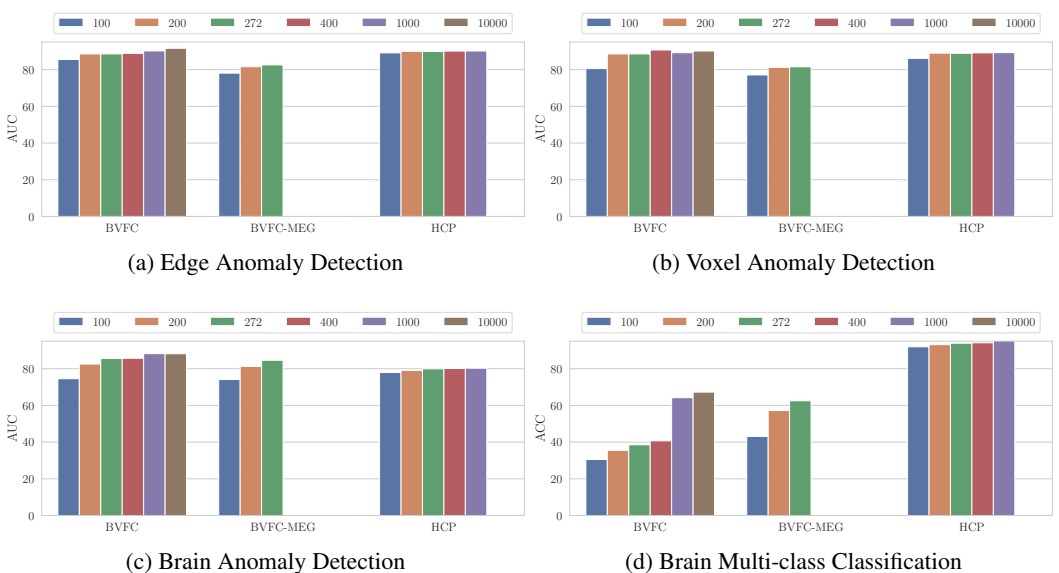

Figure 7: The effect of the number of ROIs on the performance of BRAINMIXER in different downstream tasks.

BVFC in multi-class classification task as it requires learning voxel activity (e.g., V1 and V2) not learning the higher-level aggregated visual cortex activity.

**The Effect of the Aggregation of Time Series.** Most existing studies on voxel-level brain activity aggregate the voxel activity in each time window and use a single weight relating the voxel activity to the task, called beta weight. However, aggregation of the voxel activity misses the temporal property and dynamics of voxel activity over time. To this end, we suggest using a time series encoder that

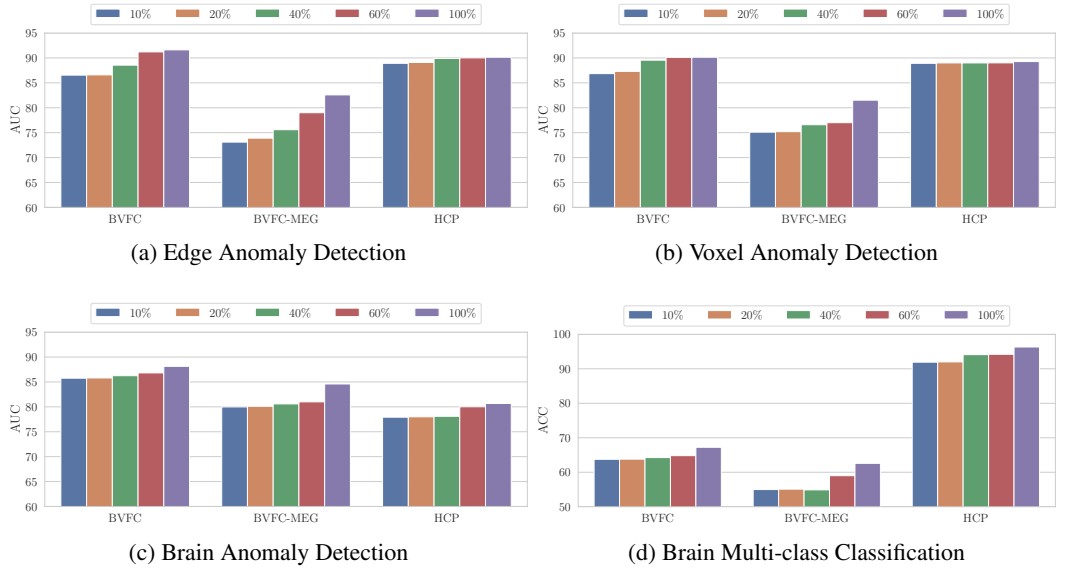

Figure 8: The effect of the aggregation of the time series on the performance of BRAINMIXER in different downstream tasks.

learn the dynamic of voxel activity over time, instead of simply aggregating them. In this experiment, we evaluate how much the aggregation of voxel activity time series can affect the performance. To this end, we take the mean of voxel activity time series and shorten it to 10%, 20%, 40%, and 60% of its original size. Figure 8 reports the results of this experiment on different downstream tasks. For datasets with low frequency sampling rate (e.g., HCP) aggregation does not significantly damage the performance. For the datasets with high frequency sampling rate (e.g., BVFC-MEG), however, aggregation of the voxel activity can significantly damage the performance ($\approx 12\%$ performance lost in the worst case and $\approx 5\%$ in the best case). These results show the importance of considering voxel activity as a time series instead of aggregating its activity and considering it as a single weight.

## F.3 The Effect of Functional and Temporal Patching

As discussed in section 3, in both VA Encoder and FC Encoder we first split the data (either time series or graph) into patches. In this section, we replace the proposed functional and temporal patching methods with some existing patching strategies as well as some baselines to evaluate their contribution in BRAINMIXER's superior performance. For patching time series data we evaluate the following methods:

1. Random Patching: We randomly group time series in multivariate time series data and treat each group as a patch.
2. Ordered Patching: We use the actual order of time series in the multivariate time series and group consecutive time series as a patch.
3. Correlation Patching: We calculate the Pearson's correlation of multivariate time series (see Equation 26) and split the data into groups base on their pairwise correlation.
4. Functional Patching: This is our designed patching method, in which we group the time series of voxels in each brain functional system as a patch.

Also, for the graph patching we evaluate the following methods:

1. 1-hop Patching: We use the 1-hop neighborhood of each node as its corresponding patch.
2. Partitioning Patching: Following He et al. [35] for graph patching, we use METIS [48], a graph clustering algorithm that partitions a graph into a pre-defined number of clusters.
3. Spectral Clustering Patching: Following Geenjaar et al. [31], we use spectral clustering patching used in this study.

Table 7: The effect of functional and temporal patching on the performance of BRAINMIXER: Mean ACC (%) $\pm$ standard deviation.

| Patching Methods | | BVFC | BVFC-MEG | HCP-Mental | HCP-Age |
|---|---|---|---|---|---|
| Time Series Patching | Random Patching | $60.42_{\pm 3.53}$ | $60.28_{\pm 1.72}$ | $91.97_{\pm 1.88}$ | $45.71_{\pm 3.69}$ |
| | Ordered Patching | $60.58_{\pm 0.60}$ | $60.55_{\pm 1.01}$ | $91.63_{\pm 1.57}$ | $47.21_{\pm 0.91}$ |
| | Correlation Patching | $66.97_{\pm 0.63}$ | $60.91_{\pm 1.46}$ | $95.08_{\pm 1.21}$ | $55.30_{\pm 0.98}$ |
| | Functional Patching | $\mathbf{67.24_{\pm 1.47}}$ | $\mathbf{62.58_{\pm 1.12}}$ | $\mathbf{96.32_{\pm 0.29}}$ | $\mathbf{57.83_{\pm 1.03}}$ |
| Graph Patching | 1-hop Patching | $63.59_{\pm 0.09}$ | $60.01_{\pm 0.18}$ | $89.97_{\pm 0.16}$ | $54.91_{\pm 0.71}$ |
| | Partitioning Patching | $66.50_{\pm 1.92}$ | $60.63_{\pm 1.87}$ | $96.14_{\pm 1.04}$ | $56.82_{\pm 1.75}$ |
| | Spectral Clustering Patching | $63.77_{\pm 0.23}$ | $59.16_{\pm 1.33}$ | $90.24_{\pm 0.95}$ | $48.34_{\pm 1.28}$ |
| | Static Random Walk Patching | $66.28_{\pm 1.52}$ | $59.94_{\pm 1.20}$ | $95.86_{\pm 0.79}$ | $57.81_{\pm 0.92}$ |
| | Functional Patching in Graph | $60.03_{\pm 0.68}$ | $54.99_{\pm 0.74}$ | $91.45_{\pm 0.80}$ | $50.11_{\pm 0.96}$ |
| | Temporal Patching | $\mathbf{67.24_{\pm 1.47}}$ | $\mathbf{62.58_{\pm 1.12}}$ | $\mathbf{96.32_{\pm 0.29}}$ | $\mathbf{57.83_{\pm 1.03}}$ |

4. Static Random Walk Patching: We replace temporal random walk in our temporal patching strategy with a static random walk. This random walk still should be able to capture structural properties but missing the dynamic of the graph.

5. Functional Patching in Graph: We use the actual brain functional systems as our patches.

6. Temporal Patching: This is our designed patching method for brain connectivity graph, in which we use temporal random walks that randomly sample temporal and structural neighborhood of each node.

In this experiments, we replace the baselines with our proposed patching method and keep the rest of the model unchanged. Results are reported in Table 7. Results show the superior performance of functional and temporal patching in time series and graph data, respectively. In time series patching, random and ordered patching perform poorly as they might group unrelated time series. Correlation patching performs better but still weaker than functional patching. The main reason for this superior performance is that we expect time series in each patch to have similar functionality and functional patching using the actual brain functional systems provides the best grouping since we know voxels in each functional system has similar functionality.

In graph patching, again our proposed temporal patching outperforms the other patching methods. Surprisingly, here functional patching performs poorly. The main reason is that in the functional connectivity graph, we connect highly correlated voxels (with respect to their activity). However, in each brain functional systems some voxels have complementary activity to others, which means while they have the same functionality, they might not have high correlation and so they are not connected. This fact results in considering disconnected subgraphs as patches, which is undesirable and so damages the performance.

### F.4 Performance Comparison using Different Metrics

We compared the performance of BRAINMIXER with baselines in section 4. In multi-class classification tasks, we used accuracy as our metrics. Also, for the anomaly detection tasks, since we have binary labels (either anomaly or normal), we used AUC-PR as the metrics. In this section, we additionally evaluate the BRAINMIXER and baselines using accuracy for anomaly detection tasks and top-1 accuracy for multi-class brain classification tasks.

**Accuracy.** In this part, we compare the performance of BRAINMIXER and baselines in anomaly detection tasks, using accuracy metric. Table 8 reports the results. Similar to Table 2, these results show the superior performance of BRAINMIXER in all edge-level, voxel-level, and brain-level anomaly detection tasks.

**Comparison of Best results.** In Table 1, we reported the average of accuracy. In this experiment, we report the best performance of each model among 20 times of running. Table 9 reports the result and show that BRAINMIXER significantly outperforms baselines.

Table 8: Performance on anomaly detection: Accuracy (%) ± standard deviation.

| | Methods | BVFC | BVFC-MEG | HCP 1% | HCP 5% | ADHD 1% | ADHD 5% | TUH-EEG 1% | TUH-EEG 5% | ASD 1% | ASD 5% |
|---|---|---|---|---|---|---|---|---|---|---|---|
| **Edge-level AD** | GOUTLIER | $62.78_{\pm2.41}$ | $55.24_{\pm2.03}$ | $60.83_{\pm1.73}$ | $58.29_{\pm1.49}$ | $63.91_{\pm1.84}$ | $61.05_{\pm2.32}$ | $64.12_{\pm1.46}$ | $61.87_{\pm1.61}$ | $59.59_{\pm0.83}$ | $56.79_{\pm1.51}$ |
| | NETWALK | $69.21_{\pm1.92}$ | $58.47_{\pm1.77}$ | $71.99_{\pm1.12}$ | $69.64_{\pm0.89}$ | $68.85_{\pm2.34}$ | $66.18_{\pm1.98}$ | $70.71_{\pm0.97}$ | $67.92_{\pm1.01}$ | $68.39_{\pm1.79}$ | $65.87_{\pm2.06}$ |
| | HYPERSAGCN | $78.84_{\pm1.73}$ | $67.47_{\pm0.92}$ | $80.33_{\pm1.61}$ | $78.61_{\pm1.24}$ | $82.05_{\pm1.75}$ | $80.57_{\pm1.32}$ | $72.45_{\pm1.67}$ | $69.15_{\pm0.89}$ | $72.73_{\pm0.92}$ | $70.68_{\pm1.77}$ |
| | GRAPHMIXER | $84.82_{\pm2.01}$ | $71.34_{\pm0.98}$ | $85.36_{\pm1.58}$ | $83.99_{\pm1.19}$ | $83.89_{\pm1.78}$ | $81.58_{\pm0.99}$ | $74.29_{\pm1.78}$ | $72.48_{\pm1.65}$ | $83.33_{\pm1.36}$ | $80.94_{\pm1.62}$ |
| | BRAINNETCNN | $77.26_{\pm0.89}$ | $68.78_{\pm1.49}$ | $78.43_{\pm0.94}$ | $75.08_{\pm1.33}$ | $79.22_{\pm1.97}$ | $76.54_{\pm1.17}$ | $72.11_{\pm1.49}$ | $68.81_{\pm1.06}$ | $70.52_{\pm1.47}$ | $68.85_{\pm1.80}$ |
| | BRAINGNN | $78.09_{\pm0.73}$ | $69.71_{\pm1.85}$ | $80.85_{\pm1.52}$ | $78.54_{\pm1.12}$ | $77.02_{\pm1.34}$ | $75.68_{\pm1.23}$ | $71.36_{\pm1.38}$ | $68.83_{\pm1.83}$ | $70.24_{\pm1.77}$ | $68.92_{\pm1.25}$ |
| | FBNETGEN | $78.35_{\pm1.98}$ | $70.44_{\pm1.58}$ | $80.82_{\pm1.25}$ | $78.30_{\pm1.56}$ | $77.66_{\pm0.89}$ | $75.74_{\pm1.78}$ | $72.11_{\pm1.57}$ | $69.63_{\pm1.47}$ | $70.97_{\pm1.42}$ | $69.69_{\pm1.18}$ |
| | ADMIRE | $84.91_{\pm1.71}$ | $72.60_{\pm1.91}$ | $85.23_{\pm1.07}$ | $83.58_{\pm1.44}$ | $84.43_{\pm1.68}$ | $82.34_{\pm1.19}$ | $74.88_{\pm1.07}$ | $72.34_{\pm1.72}$ | $85.84_{\pm1.55}$ | $82.76_{\pm1.48}$ |
| | PTGB | $84.13_{\pm1.78}$ | $71.24_{\pm1.43}$ | $84.44_{\pm1.86}$ | $83.61_{\pm1.35}$ | $84.75_{\pm1.49}$ | $82.83_{\pm1.54}$ | $73.94_{\pm1.56}$ | $71.18_{\pm1.48}$ | $85.78_{\pm1.22}$ | $82.89_{\pm1.61}$ |
| | BNTRANSFORMER | $82.52_{\pm1.64}$ | $73.21_{\pm1.78}$ | $84.92_{\pm1.29}$ | $82.85_{\pm1.61}$ | $83.52_{\pm1.81}$ | $82.72_{\pm1.31}$ | $74.07_{\pm1.36}$ | $72.54_{\pm1.15}$ | $73.76_{\pm1.80}$ | $71.18_{\pm1.59}$ |
| | **BRAINMIXER** | $\mathbf{87.75_{\pm1.58}}$ | $\mathbf{79.12_{\pm1.53}}$ | $\mathbf{88.19_{\pm1.97}}$ | $\mathbf{87.83_{\pm1.39}}$ | $\mathbf{89.77_{\pm1.12}}$ | $\mathbf{88.12_{\pm1.57}}$ | $\mathbf{79.88_{\pm1.24}}$ | $\mathbf{77.47_{\pm1.14}}$ | $\mathbf{89.92_{\pm1.57}}$ | $\mathbf{88.39_{\pm1.62}}$ |
| **Voxel-level AD** | USAD | $65.13_{\pm2.23}$ | $61.28_{\pm1.91}$ | $63.38_{\pm2.11}$ | $62.18_{\pm1.42}$ | $70.41_{\pm1.71}$ | $69.06_{\pm1.39}$ | $69.46_{\pm2.32}$ | $68.07_{\pm1.88}$ | $64.29_{\pm2.12}$ | $63.41_{\pm1.87}$ |
| | TST | $67.12_{\pm2.06}$ | $67.10_{\pm2.16}$ | $67.16_{\pm1.00}$ | $66.03_{\pm2.10}$ | $72.11_{\pm1.82}$ | $70.50_{\pm2.04}$ | $70.31_{\pm1.89}$ | $69.32_{\pm2.15}$ | $67.14_{\pm2.08}$ | $66.41_{\pm1.81}$ |
| | MVTS | N/A | N/A | N/A | N/A | N/A | N/A | $73.15_{\pm1.79}$ | $73.01_{\pm2.08}$ | N/A | N/A |
| | GOUTLIER | $61.28_{\pm1.78}$ | $59.33_{\pm1.16}$ | $61.47_{\pm1.70}$ | $61.12_{\pm1.93}$ | $66.83_{\pm1.82}$ | $64.79_{\pm2.16}$ | $62.26_{\pm2.02}$ | $61.33_{\pm1.51}$ | $57.75_{\pm1.95}$ | $56.87_{\pm1.95}$ |
| | NETWALK | $65.31_{\pm1.90}$ | $62.38_{\pm2.13}$ | $64.65_{\pm1.76}$ | $63.39_{\pm2.15}$ | $73.71_{\pm1.88}$ | $71.12_{\pm1.41}$ | $69.40_{\pm1.87}$ | $68.89_{\pm1.76}$ | $69.71_{\pm1.63}$ | $68.49_{\pm2.11}$ |
| | HYPERSAGCN | $75.01_{\pm1.50}$ | $70.01_{\pm1.72}$ | $78.61_{\pm1.98}$ | $76.24_{\pm1.46}$ | $81.22_{\pm1.54}$ | $80.34_{\pm1.26}$ | $72.28_{\pm1.45}$ | $70.27_{\pm2.20}$ | $72.59_{\pm1.56}$ | $72.42_{\pm1.92}$ |
| | GRAPHMIXER | $75.06_{\pm1.45}$ | $71.34_{\pm2.14}$ | $79.32_{\pm1.56}$ | $78.14_{\pm1.98}$ | $79.45_{\pm1.87}$ | $77.56_{\pm1.79}$ | $69.70_{\pm1.43}$ | $68.18_{\pm2.03}$ | $70.42_{\pm1.71}$ | $70.39_{\pm1.94}$ |
| | BRAINNETCNN | $77.10_{\pm1.29}$ | $72.43_{\pm2.09}$ | $80.19_{\pm1.64}$ | $79.36_{\pm1.43}$ | $80.48_{\pm1.33}$ | $78.88_{\pm1.65}$ | $70.72_{\pm1.62}$ | $69.57_{\pm1.10}$ | $71.22_{\pm1.75}$ | $70.79_{\pm2.04}$ |
| | BRAINGNN | $76.48_{\pm1.55}$ | $72.06_{\pm1.35}$ | $80.26_{\pm1.80}$ | $79.09_{\pm1.93}$ | $80.17_{\pm1.24}$ | $77.97_{\pm2.18}$ | $70.32_{\pm2.20}$ | $68.88_{\pm2.15}$ | $70.39_{\pm1.63}$ | $69.33_{\pm2.16}$ |
| | FBNETGEN | $76.29_{\pm2.02}$ | $71.50_{\pm1.96}$ | $81.41_{\pm0.91}$ | $78.18_{\pm1.86}$ | $78.66_{\pm1.47}$ | $78.67_{\pm1.48}$ | $69.85_{\pm2.06}$ | $68.38_{\pm1.97}$ | $70.50_{\pm1.59}$ | $69.44_{\pm2.08}$ |
| | PTGB | $82.26_{\pm1.37}$ | $75.59_{\pm1.42}$ | $83.25_{\pm1.51}$ | $81.80_{\pm2.02}$ | $84.51_{\pm1.32}$ | $83.29_{\pm1.39}$ | $74.73_{\pm1.02}$ | $74.07_{\pm2.13}$ | $75.59_{\pm2.04}$ | $75.12_{\pm2.12}$ |
| | BNTRANSFORMER | $82.25_{\pm1.56}$ | $74.71_{\pm2.10}$ | $84.16_{\pm1.03}$ | $81.12_{\pm1.11}$ | $84.64_{\pm1.62}$ | $83.46_{\pm2.13}$ | $74.07_{\pm1.68}$ | $74.40_{\pm1.97}$ | $74.47_{\pm1.18}$ | $73.04_{\pm1.74}$ |
| | **BRAINMIXER** | $\mathbf{87.11_{\pm1.03}}$ | $\mathbf{80.38_{\pm1.59}}$ | $\mathbf{88.01_{\pm1.22}}$ | $\mathbf{85.84_{\pm1.32}}$ | $\mathbf{88.84_{\pm1.37}}$ | $\mathbf{86.55_{\pm1.19}}$ | $\mathbf{76.98_{\pm1.42}}$ | $\mathbf{76.11_{\pm1.52}}$ | $\mathbf{87.19_{\pm1.93}}$ | $\mathbf{87.11_{\pm1.84}}$ |
| **Brain-level AD** | USAD | $69.12_{\pm1.92}$ | $60.27_{\pm1.41}$ | $64.23_{\pm1.55}$ | $64.14_{\pm2.27}$ | $79.71_{\pm2.09}$ | $78.47_{\pm2.11}$ | $68.01_{\pm2.06}$ | $67.12_{\pm1.18}$ | $70.28_{\pm1.87}$ | $67.09_{\pm1.68}$ |
| | TST | $70.94_{\pm2.26}$ | $66.03_{\pm1.94}$ | $64.61_{\pm1.78}$ | $63.18_{\pm1.69}$ | $80.43_{\pm2.01}$ | $81.51_{\pm1.21}$ | $68.82_{\pm2.03}$ | $68.09_{\pm1.79}$ | $71.47_{\pm1.64}$ | $69.42_{\pm2.08}$ |
| | MVTS | N/A | N/A | N/A | N/A | N/A | N/A | $79.24_{\pm1.56}$ | $78.39_{\pm1.80}$ | N/A | N/A |
| | NETWALK | $70.45_{\pm1.29}$ | $68.23_{\pm1.67}$ | $66.45_{\pm1.37}$ | $65.23_{\pm1.44}$ | $80.89_{\pm1.11}$ | $80.28_{\pm1.83}$ | $67.01_{\pm1.87}$ | $65.37_{\pm1.45}$ | $71.29_{\pm1.63}$ | $69.26_{\pm1.49}$ |
| | HYPERSAGCN | $78.10_{\pm1.18}$ | $75.76_{\pm1.22}$ | $69.08_{\pm1.42}$ | $69.16_{\pm1.37}$ | $83.18_{\pm1.39}$ | $84.32_{\pm1.65}$ | $71.14_{\pm1.11}$ | $71.59_{\pm1.32}$ | $75.46_{\pm1.60}$ | $72.09_{\pm1.37}$ |
| | GMM | $79.55_{\pm1.07}$ | $76.09_{\pm1.21}$ | $71.44_{\pm1.26}$ | $71.55_{\pm1.42}$ | $82.14_{\pm1.24}$ | $83.20_{\pm1.14}$ | $72.00_{\pm1.24}$ | $72.82_{\pm1.08}$ | $75.76_{\pm1.55}$ | $72.08_{\pm1.46}$ |
| | GRAPHMIXER | $80.49_{\pm1.10}$ | $76.68_{\pm1.38}$ | $72.28_{\pm1.24}$ | $72.04_{\pm1.17}$ | $83.26_{\pm1.05}$ | $83.13_{\pm1.21}$ | $72.34_{\pm1.33}$ | $72.34_{\pm1.33}$ | $75.42_{\pm1.64}$ | $73.59_{\pm1.62}$ |
| | BRAINNETCNN | $76.29_{\pm1.32}$ | $72.70_{\pm1.13}$ | $67.59_{\pm1.19}$ | $67.89_{\pm1.78}$ | $82.77_{\pm1.46}$ | $82.04_{\pm1.17}$ | $69.83_{\pm1.29}$ | $69.29_{\pm1.12}$ | $74.86_{\pm1.38}$ | $72.15_{\pm1.38}$ |
| | BRAINGNN | $77.26_{\pm1.42}$ | $74.14_{\pm1.28}$ | $69.43_{\pm1.35}$ | $69.68_{\pm1.59}$ | $81.09_{\pm1.42}$ | $81.34_{\pm1.44}$ | $68.91_{\pm1.53}$ | $67.48_{\pm1.28}$ | $74.02_{\pm1.15}$ | $71.47_{\pm1.47}$ |
| | FBNETGEN | $76.28_{\pm1.10}$ | $68.22_{\pm1.28}$ | $68.53_{\pm1.32}$ | | $81.08_{\pm1.33}$ | $82.57_{\pm1.22}$ | $68.51_{\pm1.37}$ | $67.95_{\pm1.39}$ | $74.12_{\pm1.48}$ | $71.61_{\pm1.26}$ |
| | ADMIRE | $81.54_{\pm1.22}$ | $77.27_{\pm1.16}$ | $72.92_{\pm2.06}$ | $71.34_{\pm1.59}$ | $83.35_{\pm1.16}$ | $83.19_{\pm1.43}$ | $74.32_{\pm1.31}$ | $73.11_{\pm1.45}$ | $76.62_{\pm1.76}$ | $73.88_{\pm1.53}$ |
| | PTGB | $82.01_{\pm1.51}$ | $78.12_{\pm1.25}$ | $73.53_{\pm1.64}$ | $72.62_{\pm1.26}$ | $84.42_{\pm1.47}$ | $84.62_{\pm1.17}$ | $75.81_{\pm1.24}$ | $74.19_{\pm1.42}$ | $79.98_{\pm1.58}$ | $77.29_{\pm1.46}$ |
| | BNTRANSFORMER | $81.60_{\pm1.18}$ | $78.14_{\pm1.22}$ | $72.13_{\pm1.55}$ | $72.84_{\pm1.67}$ | $84.71_{\pm1.23}$ | $84.09_{\pm1.26}$ | $75.14_{\pm1.94}$ | $74.44_{\pm1.23}$ | $78.18_{\pm1.46}$ | $73.50_{\pm1.38}$ |
| | **BRAINMIXER** | $\mathbf{86.62_{\pm1.81}}$ | $\mathbf{83.22_{\pm2.01}}$ | $\mathbf{77.92_{\pm1.86}}$ | $\mathbf{77.13_{\pm1.35}}$ | $\mathbf{88.51_{\pm1.27}}$ | $\mathbf{88.19_{\pm1.61}}$ | $\mathbf{81.22_{\pm1.59}}$ | $\mathbf{81.74_{\pm1.27}}$ | $\mathbf{88.14_{\pm1.41}}$ | $\mathbf{85.27_{\pm1.35}}$ |

Table 9: Performance on multi-class brain classification: Top-1 ACC (%) (Best results among 20 times run).

| Methods | BVFC | BVFC-MEG | HCP-Mental | HCP-Age |
|---|---|---|---|---|
| USAD | 50.48 | 51.16 | 75.05 | 40.84 |
| HYPERSAGCN | 53.41 | 53.09 | 91.94 | 49.31 |
| GMM | 54.54 | 54.72 | 92.73 | 48.99 |
| GRAPHMIXER | 54.37 | 54.27 | 92.56 | 49.43 |
| BRAINNETCNN | 50.91 | 51.68 | 85.25 | 44.30 |
| BRAINGNN | 52.29 | 52.05 | 87.40 | 44.61 |
| FBNETGEN | 51.13 | 52.32 | 86.34 | 44.61 |
| ADMIRE | 55.73 | 56.78 | 91.65 | 49.48 |
| PTGB | 57.69 | 56.73 | 93.88 | 49.90 |
| BNTRANSFORMER | 56.37 | 56.80 | 93.17 | 49.08 |
| **BRAINMIXER** | **68.67** | **63.68** | **96.63** | **58.88** |

## F.5 Graph Regression

In this section we evaluate the performance of BRAINMIXER in a regression task and compare it with baselines. In this task, we aim to predict Achenbach adult self-report (ASR) scores in HCP dataset, which are "Aggressive", "Intrusive", and "Rule-Break" scores. In this experimental setup, we use L1 loss to fine-tune the model for the downstream regression task. We also use the commonly used metric of Mean Absolute Errors (MAEs) on the prediction of these three scores.

Table 10 reports the results. In all three regression tasks, BRAINMIXER achieves the lower MAE and outperforms all the baselines.

## F.6 Qualitative Results

In this section, we report some success and failure cases of BRAINMIXER in image classification task and detecting synthetic images based on fMRI. Figure 9 (resp. Figure 10) shows four examples

Table 10: Performance on brain network regression task: MAE ↓ (the lower value is better).

| Dataset | BRAINMIXER | BNTRANSFORMER | BRAINGNN | BRAINNETCNN | GMM |
|---|---|---|---|---|---|
| HCP-AGGRESSIVE | **0.81** | 0.96 | 1.72 | 1.59 | 1.05 |
| HCP-INTRUSIVE | **0.95** | 1.09 | 1.19 | 1.27 | 1.01 |
| HCP-RULE-BREAK | **1.06** | 1.14 | 2.01 | 1.44 | 1.38 |

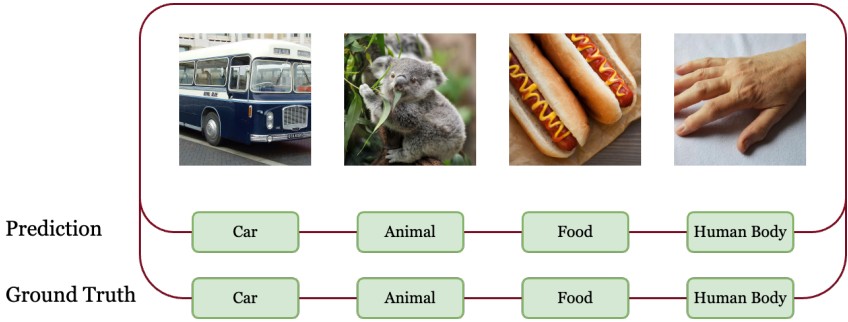

Figure 9: Examples of success cases in prediction of image labels based on fMRI.

of BRAINMIXER success (resp. failures) in prediction of the image label based on the brain activity. Finally, Figure 11 shows four example of BRAINMIXER failure in detecting synthetic (abnormal) images based on the fMRI.

## F.7 How Does BRAINMIXER Detect GAN Generated Images?

The visual cortex, responsible for processing visual information, is hierarchically organized with multiple layers building upon simpler features at lower stages [86]. Initially, neurons detect edges and colors, but on deeper levels, they specialize in recognizing more complex patterns and objects.

As we discussed in Appendix B, BVFC consists of the fMRI response when the subject sees the GAN-generated image, and we define an anomaly detection task, in which we aim to detect brain responses to GAN-generated images. While we report the performance of BRAINMIXER in detecting these anomalies, in this experiment, we examine how BRAINMIXER can detect fMRI responses to the GAN generated images. To this end, we split the test set into two groups based on BRAINMIXER's prediction: ① data samples that BRAINMIXER has detected as normal, and ② data samples that BRAINMIXER has detected as abnormal. Figure 2 (Left) reports the distribution of fMRI responses that BRAINMIXER found abnormal (i.e., corresponds to synthetic images) and Figure 2 (Right) reports the the distribution of fMRI responses that BRAINMIXER found normal (corresponds to natural images). Interestingly, while the distributions share similar patterns in lower levels (e.g., V1 and V2 voxels), higher-level voxels (e.g., V3) are less active when the subject sees non-recognizable images. This voxel activity drop in the V3 is ≈ 57%. These results are compatible with our expectation about the hierarchical structure of the visual cortex and so support that BRAINMIXER can learn a powerful representation for voxel activity.

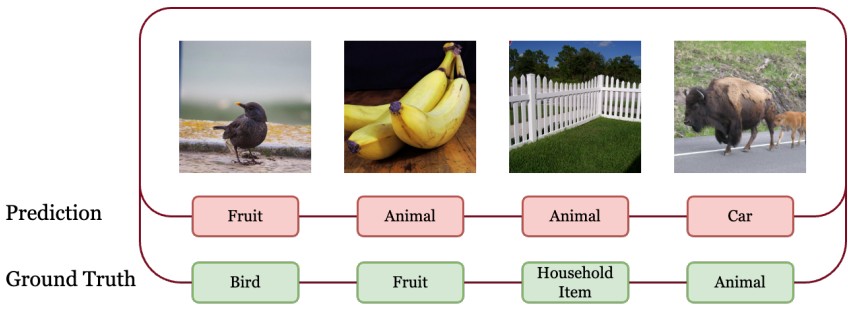

Figure 10: Examples of failure cases in prediction of image labels based on fMRI.

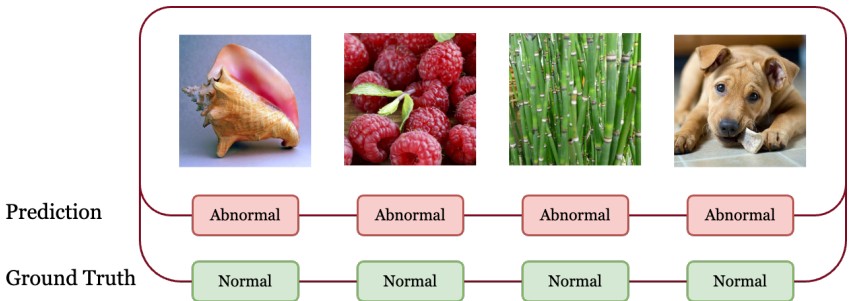

Figure 11: Examples of failure cases in detecting synthetic images based on fMRI.

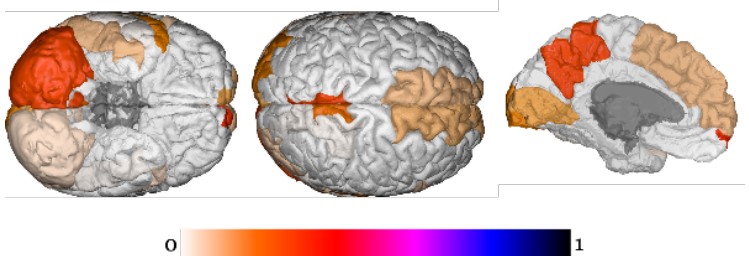

Figure 12: The distribution of detected abnormal voxels by BRAINMIXER in condition ASD group

Table 11: Performance on multi-class brain classification using different objectives (ACC).

| Methods | BVFC | BVFC-MEG | HCP-Mental | HCP-Age |
|---|---|---|---|---|
| BRAINMIXER with contrastive learning and margin-based pairwise loss | 47.30 | 49.77 | 53.78 | 49.41 |
| BRAINMIXER with DGI | 63.49 | 60.82 | 94.12 | 58.36 |
| BRAINMIXER | **68.67** | **63.68** | **96.63** | **58.88** |

## F.8   Case Study: ASD

In this experimental design we train our model on a healthy control group, which lets the model learn normal brain patterns. After the training, we test our model on the ASD group and report the abnormal brain regions in the ASD group. The most repeatedly abnromal regions in ASD group are ① Right Cerebellum Cortex, ② Right-precuneus, and ③ Left-lingual. Our findings about the abnormal activity in the cerebellum cortex is consistent with previous studies [71].

## F.9   The Effect of Objective

In this experiment, to evaluate the significance of our loss function, we train the model with two other well known loss functions. ① Contrastive learning: we replace our loss function with margin-based pairwise. In this loss function, we aim to maximize the distances of positive and negative samples. ② Deep Graph InfoMax [91]: We use the encoding of each node as its local feature. Furthermore, we use the suumary of the all encoding as the global encoding of the graph. Results are reported in Table 11.

## F.10   Number of Parameters

To compare the capacity of the models, we report the number of parameters in Table 12.

Table 12: Number of parameters in different models designed for neuroimage data.

| Methods | BVFC | BVFC-MEG | HCP-Mental | HCP-Age |
|---|---|---|---|---|
| BRAINNETCNN | 4.1M | 0.97M | 1.1M | 1.1M |
| BRAINGNN | 6.7M | 0.8M | 0.8M | 0.8M |
| FBNETGEN | 7.2M | 0.56M | 1.1M | 0.9M |
| ADMIRE | 89.3M | 4.9M | 4.2M | 4.2M |
| PTGB | 146.1M | 10.1M | 9.6M | 9.6M |
| BNTRANSFORMER | 187.8M | 8.7M | 12.4M | 12.4M |
| BRAINMIXER | 117.4M | 9.4M | 8.3M | 8.3M |

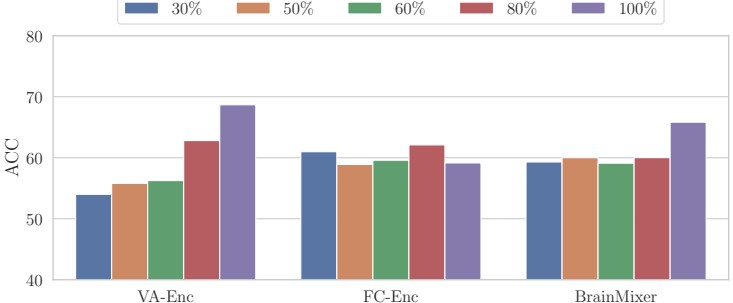

Figure 13: The performance of the BRAINMIXERwith different capacity on BVFC.

### F.11 Effect of the Number of Parameters on Accuracy

To evaluate the effect of the number of parameters on the performance of the model, we use BVFC and employ BRAINMIXER with different capacity. We restrict BRAINMIXER's and its encoders' capacity to 80%, 60%, 50%, and 30% of their original capacity, which we have reported in Table 12. The results are reported in Figure 13.

## G  Limitations and Future Work

In this work, we present an unsupervised pre-training framework, BRAINMIXER, that bridges the representation learning of voxel activity and functional connectivity by maximizing their mutual information. The promising performance of BRAINMIXER in several downstream tasks raises many interesting directions for future studies: While BRAINMIXER with a simple MLP can successfully classify observed images based on fMRI, one future direction is to pair BRAINMIXER with diffusion models [96] to directly decode brain visual system in an end-to-end manner. There are, however, a few limitations for BRAINMIXER: ⓘ In this study, we focus on designing a powerful unsupervised framework that could provide us with a robust and effective brain activity representation. However, reliability of machine learning methods for downstream tasks in sensitive domains (like healthcare) is critical. Evaluation of BRAINMIXER's prediction reliability and modifying BRAINMIXER so that it can provide us with the uncertainty of its prediction is left for future studies. ⓘⓘ The current approach is capable of using one neuroimage modalities, while different neuroimage modalities can provide complementary information, which can help understanding and detecting neurological disease or disorders. One potential future work is to design multimodal BRAINMIXER, where it can learn from different neuroimage modalities, taking advantage of their complementary information.

