# OpenReview forum: "Unsupervised Representation Learning of Brain Activity via Bridging Voxel Activity and Functional Connectivity"
_NeurIPS.cc/2023/Workshop/AI4Science — NeurIPS2023-AI4Science Poster_

### Official Review · Reviewer_6Jc5 · 2023-10-12
**Very interesting, complete work, with good results and presentation**

**Rating:** 8
**Confidence:** 4

**Review:**

This work targets a gap in the neuroimaging literature regarding architectures that consider both the voxel- and ROI-level information. This is generally an underexplored subfield and therefore I believe it perfectly fits the objectives of workshop.

I think this is a really good piece of work for a workshop. The information is well presented in the context of the broader literature, and the results show systematic better performance across a wide range of datasets and tasks. On top of that, there's a useful ablation analysis to better understand the contributions of the different components. For all these reasons, this is a clear accept and I look forward to see the complete paper so I can see more details in Appendix, which for some reason was not available at this stage.

In terms of cons of this work, I think there are important parts that are confusing and not clear, as follows:
1. How are the ROIs defined for the different tasks and datasets? I believe this is what the paper defines as "functional systems". How could different parcellations influence these results? I believe that for some baselines, which are directly dependent on the parcellation created, the results could change a lot.
2. From the notation set in section 3, it seems that the connections between voxels are different from the correlation matrix between all voxels. In this sense, what exactly is a "connection" between two voxels, that comes with the chosen representation of the data itself, and is therefore something that needs to be decided a prior?
3. I find the choice of showing only the accuracy metric for table 1 a bit of cherry-picking. It is well-known that for a lot of (binary) classification tasks, accuracy is very limited; therefore, I think that if the authors had to choose only one metric given limited space, they should have gone with something else like F1 score or, at least, mention whether the results are similar for other metrics. It is then not clear whether the "outstanding performance" in the brain classification task is just for accuracy, which is not the most useful metric to truly evaluate this task.
4. How much of this improved performance is due to increased architectural complexity or hyperparameter search? For example, indicating the number of parameters and training/inference times would be useful to understand this, maybe in appendix.
5. In introduction, the paper mentions two works which I was not aware before: 50 and 52. The way they are presented makes me think that they are the only two in the literature more similar to this work in the sense that they tackle the same gap in the literature (ie, lack of methods that target more than one scale). Therefore, I'd have expected that these two works should have been included in the baseline comparisons and the related work section.
6. There's a typo in line 182: "minuetes" instead of "minutes"

---

### Official Review · Reviewer_wEx9 · 2023-10-25
**a novel encoding method based on both functional connectivity and voxel-wise activity**

**Rating:** 7
**Confidence:** 4

**Review:**

The authors proposed an attention-based method to produce embeddings based on both ROI-based functional connectivity and voxel-wise activity for neuroimaging data. Overall, it is a nicely-written paper. The motivation is clearly expressed, and experiments which include comparison with baselines and ablation studies are solid.

With that said, the paper can be improved if the authors address the following issues

1. section 3.1, line 134 “There does not exist in general a canonical grid of the brain to encode voxel activities, which makes patch extraction challenging”. I’m not sure what the first half means. it is a standard preprocessing procedure to project individual brain to a common space (e.g. MNI152 for volumes / grayordinate for surfaces) and therefore there is actually a "canonical grid'. Besides, I’m confused about what “patch extraction” is in the second half. I don’t think it’s a standard terminology used in fMRI studies. Based on the following context, it may stand for extracting the voxel activities from different ROIs. However, if that is the case, it is actually not challenging in a common space like grayordinate.

2. Functional patching (line 146)
        1. is the X’s temporal dimension a typo?
        2. the whole brain spatiotemporal data is split to different ROIs in this paragraph, but it seems to be irrelevant to the next steps (voxel-mixer still takes the whole X)? this is confusing

---

### Meta-Review · Area_Chair_475B · 2023-10-26

**Recommendation:** Accept (Poster)
**Confidence:** 4

**Metareview:**

SUMMARY: Both reviewers appreciated this paper for the sound problem formulation, methodology, and design of experiments and lean towards acceptance. There are still pending issues with the manuscript: such as lack of clarity on datasets and pre-processing, as well as choice of baselines and evaluation metrics. A major issue with the paper is that the appendix is referred to several times in the text (when detailed explanations are required), but has not been included with the manuscript submission or supplementary.

POINTS TO ADDRESS:

1. Please clarify the parcellation scheme used as this can have a drastic effect on performance
2. Please include the appendix with details on the datasets, class prevalence for classification
3. Please include a justification on the choice of only accuracy as a metric or provide extended table of results.
4. Please include a clarification on the choice of baselines, specifically why [50] and [51], which solve the same problem as the paper compare.
5. Preferably include a discussion on relative model sizes and training time.
6. Please clarify the description in Section 3.1 with the Figure, as pointed out by Reviewer 6Jc5, this is ambiguous.